# Pre-training Protein Structure Encoder via Siamese Diffusion Trajectory Prediction

## Abstract

Inspired by the governing role of protein structures on protein functions, structure-based protein representation pre-training is recently gaining interest but remains largely unexplored. Along this direction, pre-training by maximizing the mutual information (MI) between different descriptions of the same protein (*i.e.*, correlated views) has shown some preliminary promise, while more in-depth studies are required to design more informative correlated views. Previous view designs focus on capturing structural motif co-occurrence in the same protein structure, while they cannot capture detailed atom- and residue-level interactions and also the statistical dependencies of residue types along protein sequences. To address these limitations, we propose the Siamese Diffusion Trajectory Prediction (**SiamDiff**) method. SiamDiff employs the **multimodal diffusion process** as a faithful simulation of the structure-sequence co-diffusion trajectory that gradually and smoothly approaches the folded structure and the corresponding sequence of a protein from scratch. Upon a native protein and its correlated counterpart obtained with random structure perturbation, we build two multimodal (structure and sequence) diffusion trajectories and regard them as two correlated views. A principled theoretical framework is designed to maximize the MI between such paired views, such that the model can acquire the atom- and residue-level interactions underlying protein structural changes and also the residue type dependencies. We study the effectiveness of SiamDiff on both residue-level and atom-level structural representations. Experimental results on EC and ATOM3D benchmarks show that the performance of SiamDiff is consistently competitive on all benchmark tasks, compared with existing baselines. The source code will be made public upon acceptance.

## 1 Introduction

In the past year, thanks to the rise of highly accurate and efficient protein structure predictors based on deep learning (Jumper et al., 2021; Baek et al., 2021), the gap between the number of reported protein sequences and corresponding (computationally) solved structures is greatly narrowed. These advances open the opportunity of self-supervised protein representation learning based on protein structures, *i.e.*, learning informative protein representations from massive protein structures without using any annotation. Compared to extensively studied sequence-based protein self-supervised learning like protein language models (Elnaggar et al., 2021; Rives et al., 2021), structure-based methods could learn more effective representations to indicate protein functions, since a protein sequence determines its structure, and the structure is the determinant of its diverse functions (Harms & Thornton, 2010).

To attain this goal, some recent works have explored different self-supervised learning strategies on protein structures, including contrastive learning (Zhang et al., 2022; Hermosilla & Ropinski, 2022), self-prediction (Zhang et al., 2022; Chen et al., 2022) and denoising score matching (Guo et al., 2022; Wu et al., 2022a). Among these works, mutual information (MI) maximization based methods (Zhang et al., 2022; Hermosilla & Ropinski, 2022) achieve superior performance on protein function and structural class prediction. At the core of these methods, different structural descriptions of the same protein (*i.e.*, correlated views) are built to capture the co-occurrence of structural motifs. However, such view construction scheme fails to capture detailed atom- and residue-level interactions and also the statistical dependencies of residue types along protein sequences. Therefore, the MI maximization based on these views may not produce effective representations for the tasks that require to model detailed local structures (*e.g.*, the Residue Identity task from ATOM3D (Townshend et al., 2020)) or

minor differences on structures and sequences due to point mutations (*e.g.*, the Mutation Stability Prediction task from ATOM3D), as demonstrated by the experimental results in Tables 1 and 2.

To tackle such limitation, in this work, we propose the Siamese Diffusion Trajectory Prediction (**SiamDiff**) method to jointly model fine-grained atom- and residue-level interactions and residue type dependencies. Specifically, given a native protein, we first derive its correlated counterpart by random structure perturbation. We further extend the original protein and the generated counterpart respectively by the *multimodal diffusion process*, in which we transform both the protein structure and the protein sequence towards random distribution by gradually and smoothly adding noises. Such a diffusion process is verified as a faithful simulation of the structure-sequence co-diffusion trajectory by recent studies (Anand & Achim, 2022; Luo et al., 2022). We regard the diffusion trajectories of the original protein and the generated counterpart as two correlated views, and a principled theoretical framework is designed to maximize the MI between such paired views. Under such a learning framework, the model can acquire the atom- and residue-level interactions underlying protein structural changes and also the residue type dependencies. The learned protein representations are expected to boost diverse types of downstream tasks.

SiamDiff can be flexibly applied to both residue-level and atom-level structures for effective representation learning. We employ different self-supervised algorithms to pre-train residue-level and atom-level structure encoders, and the pre-trained models are extensively evaluated on Enzyme Commission number prediction (Gligorijević et al., 2021) and ATOM3D (Townshend et al., 2020) benchmarks. Experimental results verify that SiamDiff can consistently achieve competitive performance on all benchmark tasks and on both structure levels, in contrast to existing baselines.

## 2    RELATED WORK

**Protein Structure Representation Learning.**    The community witnessed a surge of research interests in learning informative protein structure representations using structure-based encoders and training algorithms. The encoders are designed to capture protein structural information on different granularity, including residue-level structures (Gligorijević et al., 2021; Wang et al., 2022b; Zhang et al., 2022), atom-level structures (Hermosilla et al., 2021; Jing et al., 2021a; Wang et al., 2022a) and protein surfaces (Gainza et al., 2020; Sverrisson et al., 2021; Somnath et al., 2021).

Recent works study pre-training on massive unlabeled protein structures for generalizable representations, covering contrastive learning (Zhang et al., 2022; Hermosilla & Ropinski, 2022), self-prediction of geometric quantities (Zhang et al., 2022; Chen et al., 2022) and denoising score matching (Guo et al., 2022; Wu et al., 2022a). All these methods only employ native proteins for pre-training. By comparison, the proposed SiamDiff uses the information from multimodal diffusion trajectories to better acquire atom- and residue-level interactions and residue type dependencies.

**Diffusion Probabilistic Models (DPMs).** DPM was first proposed in Sohl-Dickstein et al. (2015) and has been recently rekindled for its strong performance on image and waveform generation (Ho et al., 2020; Chen et al., 2020a). Recent works (Nichol & Dhariwal, 2021; Song et al., 2020; 2021) have improved training and sampling for DPMs. Besides the DPMs for continuous data, some works study discrete DPMs and achieve impressive results on generating texts (Austin et al., 2021; Li et al., 2022), images (Austin et al., 2021) and image segmentation data (Hoogeboom et al., 2021).

Inspired by these progresses, DPMs have been recently adopted to solve the problems in chemistry and biology domain, including molecule generation (Xu et al., 2022; Hoogeboom et al., 2022; Wu et al., 2022b), molecular representation learning (Liu et al., 2022), protein design (Anand & Achim, 2022; Luo et al., 2022) and motif-scaffolding (Trippe et al., 2022). In this work, we novelly study how DPMs can help protein representation learning, which aligns with a recent effort (Abstreiter et al., 2021) on diffusion-based image representation learning.

## 3    PRELIMINARIES

### 3.1    PROBLEM DEFINITION

**Notations.**    A protein with $n_r$ residues (amino acids) and $n_a$ atoms can be represented as a sequence-structure tuple $\mathcal{P} = (\mathcal{S}, \mathcal{R})$. We use $\mathcal{S} = [s_1, s_2, \cdots, s_{n_r}]$ to denote its sequence with $s_i$ as the type of the $i$-th residue, while $\mathcal{R} = [\boldsymbol{r}_1, \boldsymbol{r}_2 ..., \boldsymbol{r}_{n_a}] \in \mathbb{R}^{n_a \times 3}$ denotes its structure with $\boldsymbol{r}_i$ as the Cartesian

coordinates of the $i$-th atom. In the paper, we construct a graph for each protein with edges connecting atoms with the Euclidean distance lower than a threshold. Besides, we also consider residue-level protein graphs, a concise version of atom graphs that enable efficient message passing between nodes and edges. As in Zhang et al. (2022), we only keep the alpha carbon atom of each residue and add sequential, radius and K-nearest neighbor edges as different types of edges.

**Equivariance.** *Equivariance* is ubiquitous in machine learning for modeling the symmetry in physical systems (Thomas et al., 2018; Weiler et al., 2018; Batzner et al., 2022) and is shown to be critical for successful design and better generalization of 3D neural networks (Köhler et al., 2020). Formally, a function $\mathcal{F} : \mathcal{X} \to \mathcal{Y}$ is equivariant *w.r.t.* a group $G$ if $\mathcal{F} \circ \rho_{\mathcal{X}}(x) = \rho_{\mathcal{Y}} \circ \mathcal{F}(x)$, where $\rho_{\mathcal{X}}$ and $\rho_{\mathcal{Y}}$ are transformations corresponding to an element $g \in G$ acting on $\mathcal{X}$ and $\mathcal{Y}$, respectively. The function is invariant *w.r.t* $G$ if the transformations $\rho_{\mathcal{Y}}$ is identity. In this paper, we consider the SE(3) group, *i.e.*, rotations and translations in 3D space.

**Problem Definition.** Given a set of unlabeled proteins $\mathbb{P} = \{\mathcal{P}_1, \mathcal{P}_2, ...\}$, our goal is to train a protein encoder $\phi(\mathcal{S}, \mathcal{R})$ to extract representations that are SE(3)-invariant *w.r.t.* protein structures $\mathcal{R}$.

## 3.2 Diffusion Models on Proteins

Diffusion models are a class of deep generative models with latent variables encoded by a *forward diffusion process* and decoded by *a reverse generative process* (Sohl-Dickstein et al., 2015). There have been recent efforts on applying denoising diffusion models for protein generation (Luo et al., 2022; Anand & Achim, 2022). We use $\mathcal{P}^0$ to denote the ground-truth protein and $\mathcal{P}^t$ for $t = 1, \cdots, T$ to be the latent variables over $T$ diffusion steps. Modeling the protein as an evolving thermodynamic system, the forward process gradually injects small noise to the data $\mathcal{P}^0$ until reaching a random noise distribution at time $T$. The reverse process learns to denoise the latent variable towards the data distribution. Both processes are defined as Markov chains:

$$q(\mathcal{P}^{1:T}|\mathcal{P}^0) = \prod_{t=1}^{T} q(\mathcal{P}^t|\mathcal{P}^{t-1}), \;\; p_\theta(\mathcal{P}^{0:T-1}|\mathcal{P}^T) = \prod_{t=1}^{T} p_\theta(\mathcal{P}^{t-1}|\mathcal{P}^t), \tag{1}$$

where $q(\mathcal{P}^t|\mathcal{P}^{t-1})$ defines the forward process at step $t$ and $p_\theta(\mathcal{P}^{t-1}|\mathcal{P}^t)$ with learnable parameters $\theta$ defines the reverse process at step $t$.

The generation of a protein relies on the joint diffusion process on structures and sequences. Following Luo et al. (2022), we assume 1) the separate definition of the forward process on structures and sequences and 2) the conditional independence of sequences and structures in the reverse process:

$$q(\mathcal{P}^t|\mathcal{P}^{t-1}) = q(\mathcal{R}^t|\mathcal{R}^{t-1}) \cdot q(\mathcal{S}^t|\mathcal{S}^{t-1}), \;\; p_\theta(\mathcal{P}^{t-1}|\mathcal{P}^t) = p_\theta(\mathcal{R}^{t-1}|\mathcal{P}^t) \cdot p_\theta(\mathcal{S}^{t-1}|\mathcal{P}^t). \tag{2}$$

Next, we discuss how to define the diffusion models on protein structures and sequences, respectively.

**Diffusion models on 3D structures.** Since the coordinates of atoms are continuous variables in the 3D space, the forward process can be defined by adding random Gaussian noise. Then, the reverse process can be parameterized as a Gaussian with a learnable mean and user-defined variance. That is,

$$q(\mathcal{R}^t|\mathcal{R}^{t-1}) = \mathcal{N}(\mathcal{R}^t; \sqrt{1-\beta_t}\mathcal{R}^{t-1}, \beta_t I), \;\; p_\theta(\mathcal{R}^{t-1}|\mathcal{P}^t) = \mathcal{N}(\mathcal{R}^{t-1}; \mu_\theta(\mathcal{P}^t, t), \sigma_t^2 I), \tag{3}$$

where $\beta_1, ..., \beta_T$ are a series of fixed variances and $\sigma_t$ can be any user-defined variance. Since $\mathcal{R}^t$ is available as an input, following Ho et al. (2020), we reparameterize the mean $\mu_\theta(\mathcal{P}^t, t)$ as:

$$\mu_\theta(\mathcal{P}^t, t) = \frac{1}{\sqrt{\alpha_t}} \left( \mathcal{R}^t - \frac{\beta_t}{\sqrt{1-\bar{\alpha}_t}} \epsilon_\theta(\mathcal{P}^t, t) \right), \tag{4}$$

where $\alpha_t = 1 - \beta_t$, $\bar{\alpha}_t = \prod_{s=1}^{t} \alpha_s$ and the neural network $\epsilon_\theta(\cdot)$ learns to decorrupt the data and should be translation-invariant and rotation-equivariant *w.r.t.* the protein structure $\mathcal{R}^t$.

**Diffusion models on sequences.** The key of diffusion models on discrete attributes is to define transition matrices for the Markov chain. Austin et al. (2021) adds an absorbing state [MASK] for the Markov chain and then each residue either stays the same or transits to [MASK] with some probability at each step. For the reverse process, a neural network $\tilde{p}_\theta$ is defined to predict the probability of $\mathcal{S}^0$ and then parameterize the diffusion trajectory with the probability $q(\mathcal{S}^{t-1}|\mathcal{S}^t, \tilde{\mathcal{S}}^0)$. That is,

$$q(\mathcal{S}^t|\mathcal{S}^{t-1}) = \text{Cat}\left( \mathcal{S}^t; \boldsymbol{p} = \boldsymbol{s}^{t-1}\boldsymbol{Q}^t \right), \;\; p_\theta(\mathcal{S}^{t-1}|\mathcal{P}^t) \propto \sum_{\tilde{\mathcal{S}}^0} q(\mathcal{S}^{t-1}|\mathcal{S}^t, \tilde{\mathcal{S}}^0) \cdot \tilde{p}_\theta(\tilde{\mathcal{S}}^0|\mathcal{P}^t), \tag{5}$$

where $\boldsymbol{s}^t \in \mathbb{R}^{n_r \times 21}$ denotes the one-hot feature for the sequence $\mathcal{S}^t$, $\boldsymbol{Q}^t \in \mathbb{R}^{21 \times 21}$ denotes the corresponding transition matrix at step $t$, $\text{Cat}(\cdot)$ is the categorical distribution and $\tilde{\mathcal{S}}^0$ enumerates all possible residue types.

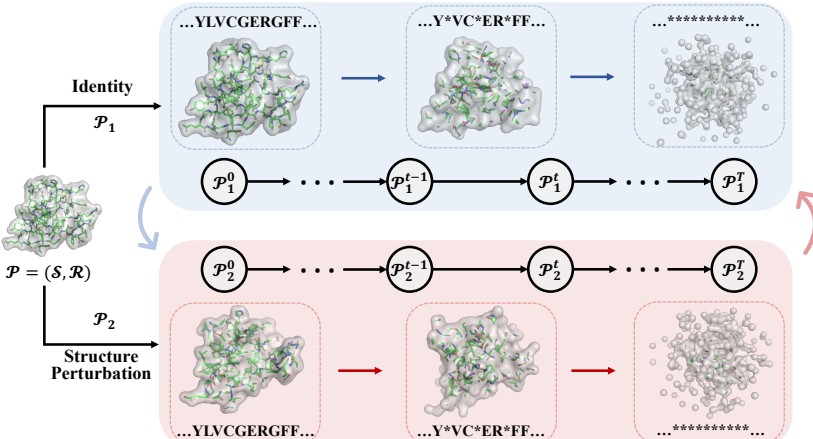

Figure 1: High-level illustration of SiamDiff. Given a protein $\mathcal{P} = (\mathcal{S}, \mathcal{R})$, we first obtain two correlated starting states $\mathcal{P}_1$ and $\mathcal{P}_2$ via identity and structure perturbation functions, respectively. Then, we apply a multimodal diffusion process on the starting states to construct two siamese trajectories, where we use * to denote masked residues. To maximize the mutual information between these trajectories, we predict the noises on one trajectory with representations from the other.

# 4 SIAMDIFF: SIAMESE DIFFUSION TRAJECTORY PREDICTION

## 4.1 MOTIVATION AND OVERVIEW

**Motivation.** Recent self-supervised learning approaches learn informative representations by maximizing mutual information (MI) between representations of multiple views of the same object with shared information, whose effectiveness has been proved on images (Chen et al., 2020b), natural languages (Hjelm et al., 2018), graphs (Hassani & Khasahmadi, 2020), small molecules (Liu et al., 2021) and proteins (Zhang et al., 2022). For MI maximization based protein representation learning, previous works (Zhang et al., 2022; Hermosilla & Ropinski, 2022) only focus on capturing protein substructure co-occurrences, while they fail to capture two important properties of proteins: (1) the detailed atom- and residue-level interactions, and (2) the statistical dependencies of residue types along protein sequences.

In this work, we introduce the *multimodal diffusion trajectory* (*i.e.*, the diffusion processes of a pair of corresponding sequence and structure) to incorporate these two properties, where (1) rich atom and residue interactions along protein formation can be learned from the structure diffusion process, and (2) residue type dependencies can be learned from the sequence diffusion process. By maximizing the MI between the multimodal diffusion trajectories of a pair of correlated proteins, more informative protein representations can be learned. The graphical illustration of this idea is shown in Fig. 1. We next present the overview of two key components of our learning framework.

**Correlated protein diffusion trajectories construction.** Our framework is based on the contrast of correlated protein trajectories, *a.k.a.*, siamese trajectories. These trajectories are generated by a *multimodal diffusion process*, which is verified to be effective for protein structure-sequence co-design (Anand & Achim, 2022; Luo et al., 2022) and thus can reflect physical and chemical principles underlying protein formation (Anfinsen, 1972; Dill & MacCallum, 2012).

Specifically, we first construct two correlated states from one protein as the starting points of the trajectories. Given the original protein $\mathcal{P} = (\mathcal{S}, \mathcal{R})$, we deem it as the native state $\mathcal{P}_1 = (\mathcal{S}_1, \mathcal{R}_1)$ and build a correlated state $\mathcal{P}_2 = (\mathcal{S}_2, \mathcal{R}_2)$ by randomly perturbing the protein structure, *i.e.*, $\mathcal{S}_2 = \mathcal{S}_1, \mathcal{R}_2 = \mathcal{R}_1 + \epsilon$, where $\epsilon \in \mathbb{R}^{n_a \times 3}$ is the noise drawn from a normal distribution.

Then, for states $\mathcal{P}_1$ and $\mathcal{P}_2$, we sample their diffusion trajectories $\mathcal{P}_1^{0:T}$ and $\mathcal{P}_2^{0:T}$ with the multimodal diffusion process defined in Sec. 3.2. Take $\mathcal{P}_1 = (\mathcal{S}_1, \mathcal{R}_1)$ for example. We start the diffusion process from the state $\mathcal{P}_1^0 = \mathcal{P}_1$. The diffusion process is defined by the joint diffusion on structures and sequences, *i.e.*, $q(\mathcal{P}_1^{1:T} | \mathcal{P}_1^0) = q(\mathcal{R}_1^{1:T} | \mathcal{R}_1^0) q(\mathcal{S}_1^{1:T} | \mathcal{S}_1^0)$. We utilize the conditional Gaussian

distributions in (3) to derive trajectories on structures $\mathcal{R}_1^{1:T}$ and the discrete Markov chain in (5) to derive the sequence diffusion processes $\mathcal{S}_1^{1:T}$. In this way, we define the trajectory $\mathcal{P}_1^{0:T} = \{(\mathcal{S}_1^t, \mathcal{R}_1^t)\}_{t=0}^T$ for $\mathcal{P}_1$ and can derive the siamese trajectory $\mathcal{P}_2^{0:T} = \{(\mathcal{S}_2^t, \mathcal{R}_2^t)\}_{t=0}^T$ similarly.

**Maximizing mutual information (MI) between representations from siamese trajectories.** The main focus of our approach is to maximize the MI between representations of siamese trajectories constructed as above (see App. A for related works about MI maximization). For notations, we use the bold symbol to denote the representation of an object and use $\boldsymbol{P}_1^{0:T}$ and $\boldsymbol{P}_2^{0:T}$ to denote the corresponding random variables of representations of the siamese trajectories $\boldsymbol{\mathcal{P}}_1^{0:T}$ and $\boldsymbol{\mathcal{P}}_2^{0:T}$. Because directly optimizing MI is intractable, we instead maximize an approximate lower bound of MI described in the following proposition (see proof in App. B.1).

**Proposition 1** *With some approximations, the mutual information between representations of two siamese trajectories is lower bounded by:*

$$I(\boldsymbol{P}_1^{0:T}; \boldsymbol{P}_2^{0:T}) \geq -\frac{1}{2}(\mathcal{L}^{(2\to1)} + \mathcal{L}^{(1\to2)}) + C,$$

*where $C$ is a constant independent of our encoder and the term from trajectory $\mathcal{P}_b^{0:T}$ to $\mathcal{P}_a^{0:T}$ is defined as*

$$\mathcal{L}^{(b\to a)} := \mathbb{E}_{\mathcal{P}_a^{0:T}, \mathcal{P}_b^{0:T}} \left[ \sum_{t=1}^T D_{KL}\left( q(\mathcal{P}_a^{t-1}|\mathcal{P}_a^t, \mathcal{P}_a^0) \| p(\mathcal{P}_a^{t-1}|\mathcal{P}_a^t, \boldsymbol{\mathcal{P}}_b^{0:T}) \right) \right], \qquad (6)$$

*with $b \to a$ being either $2 \to 1$ or $1 \to 2$.*

The two terms share the similar formula as the ELBO loss in diffusion models (Ho et al., 2020). Take $\mathcal{L}^{(2\to1)}$ for example. Here $q(\mathcal{P}_1^{t-1}|\mathcal{P}_1^t, \mathcal{P}_1^0)$ is a posterior analytically tractable with our definition of each diffusion step $q(\mathcal{P}_1^t|\mathcal{P}_1^{t-1})$ in (3) and (5). The reverse process is learnt to generate a less noisy state $\mathcal{P}_1^{t-1}$ given the current state $\mathcal{P}_1^t$ and representations of the siamese trajectory $\boldsymbol{\mathcal{P}}_2^{0:T}$, which are extracted by the protein encoder to be pre-trained.

Essentially, we perform mutual prediction between two siamese trajectories, which is similar to the idea of mutual representation reconstruction in Grill et al. (2020); Chen & He (2021). In practice, though adding structural noises, $\mathcal{P}_1$ and $\mathcal{P}_2$ share information about the same protein and thus the whole trajectory of $\mathcal{P}_2$ may give too many clues for the denoising towards $\mathcal{P}_1^{t-1}$, which is harmful for pre-training. To address this issue, we parameterize $p(\mathcal{P}_1^{t-1}|\mathcal{P}_1^t, \boldsymbol{\mathcal{P}}_2^{0:T})$ with $p_\theta(\mathcal{P}_1^{t-1}|\mathcal{P}_1^t, \boldsymbol{\mathcal{P}}_2^t)$. For diffusion on sequences, we further guarantee that the same set of residues are masked in $\mathcal{S}_1^t$ and $\mathcal{S}_2^t$ to avoid the leakage of ground-truth residue types across views.

Below we describe the parameterization of the generation process $p_\theta(\mathcal{P}_1^{t-1}|\mathcal{P}_1^t, \boldsymbol{\mathcal{P}}_2^t)$ in Sec. 4.2, derive the pre-training objective from (6) in Sec. 4.3, and discuss advantages of our method and its connection with existing works in Sec. 4.4.

## 4.2 MODELING PROTEIN GENERATION PROCESS

Different from the generation process in traditional diffusion models, the parameterization of $p_\theta(\mathcal{P}_1^{t-1}|\mathcal{P}_1^t, \boldsymbol{\mathcal{P}}_2^t)$ should inject information from $\mathcal{P}_2^t$. Therefore, we apply an SE(3)-invariant encoder $\phi(\cdot)$ to $\mathcal{P}_2^t$, and the extracted atom and residue representations (denoted as $\boldsymbol{a}_2^t$ and $\boldsymbol{h}_2^t$) are employed for this denoising step. Given the conditional independence in (2), this generation process can be decomposed into that on protein structures and sequences as shown below.

**Generation process on protein structures.** As in (4), modeling the generation process of protein structures is to model the noise on $\mathcal{R}_1^t$ and gradually decorrupt the noisy structure. This can be parameterized with a noise prediction network $\epsilon_\theta(\mathcal{P}_1^t, \boldsymbol{\mathcal{P}}_2^t, t)$ that is translation-invariant and rotation-equivariant *w.r.t.* $\mathcal{R}_1^t$. Besides, the noise applied on $\mathcal{R}_1^t$ should not change with transformations on $\mathcal{R}_2^t$, so $\epsilon_\theta$ should be SE(3)-invariant *w.r.t.* $\mathcal{R}_2^t$.

To achieve these goals, we define our noise prediction network with atom representations $\boldsymbol{a}_2^t$ (which is SE(3)-invariant *w.r.t.* $\mathcal{R}_2^t$) and atom coordinates $\boldsymbol{r}_1^t$ (which is SE(3)-equivariant *w.r.t.* $\mathcal{R}_1^t$). We draw inspirations from recent works (Satorras et al., 2021) to build an equivariant output based on normalized directional vectors between adjacent atom pairs. Each edge $(i, j)$ is encoded by its length

$\|\boldsymbol{r}_{1i}^t - \boldsymbol{r}_{1j}^t\|_2$ and the representations of two end nodes $\boldsymbol{a}_{2i}^t$, $\boldsymbol{a}_{2j}^t$, and the encoded score $m_{i,j}$ will be further used for aggregating directional vectors. Specifically, we have

$$[\epsilon_\theta(\mathcal{P}_1^t, \boldsymbol{\mathcal{P}}_2^t, t)]_i = \sum_{j \in \mathcal{N}_1^t(i)} m_{i,j} \cdot \frac{\boldsymbol{r}_{1i}^t - \boldsymbol{r}_{1j}^t}{\|\boldsymbol{r}_{1i}^t - \boldsymbol{r}_{1j}^t\|_2}, \text{ with } m_{i,j} = \text{MLP}(\boldsymbol{a}_{2i}^t, \boldsymbol{a}_{2j}^t, \text{MLP}(\|\boldsymbol{r}_{1i}^t - \boldsymbol{r}_{1j}^t\|_2)),$$

where $\mathcal{N}_1^t(i)$ denotes the neighbors of the atom $i$ in the corresponding graph of $\mathcal{P}_1^t$. Note that $\epsilon_\theta(\mathcal{P}_1^t, \boldsymbol{\mathcal{P}}_2^t, t)$ achieves the equivariance requirement, as $m_{i,j}$ is SE(3)-invariant *w.r.t.* $\mathcal{R}_1^t$ and $\mathcal{R}_2^t$ while $\boldsymbol{r}_{1i}^t - \boldsymbol{r}_{1j}^t$ is translation-invariant and rotation-equivariant *w.r.t.* $\mathcal{R}_1^t$.

**Generation process on protein sequences.** As in (5), the generation process on sequences aims to predict masked residue types in $\mathcal{S}_1^0$ with a predictor $\tilde{p}_\theta$. In our setting of mutual prediction, we define the predictor based on representations of the same residues in $\mathcal{S}_2^t$, which are also masked. Hence, for each masked residue $i$ in $\mathcal{S}_2^t$, we feed its representation $\boldsymbol{h}_{2i}^t$ to an MLP and predict the type of the corresponding residue type $s_{1i}^0$ in $\mathcal{S}_1^0$:

$$\tilde{p}_\theta(\mathcal{S}_1^0 | \mathcal{P}_1^t, \boldsymbol{\mathcal{P}}_2^t) = \prod_i \tilde{p}_\theta(s_{1i}^0 | \mathcal{P}_1^t, \boldsymbol{\mathcal{P}}_2^t) = \prod_i \text{Softmax}(s_{1i}^0 | \text{MLP}(\boldsymbol{h}_{2i}^t)),$$

where the softmax function is applied over all residue types.

### 4.3 PRE-TRAINING OBJECTIVE

Given the defined forward and reverse process on two trajectories, we now derive the pre-training objective based on the mutual diffusion loss in (6). We take the term $\mathcal{L}^{(2 \to 1)}$ for example and its counterpart can be derived in the same way. It can be proved that with the independence assumptions in (2), the objective can be decomposed into a structure loss $\mathcal{L}^{(\mathcal{R}, 2 \to 1)}$ and a sequence loss $\mathcal{L}^{(\mathcal{S}, 2 \to 1)}$ (see proof in App. B.2):

$$\mathcal{L}^{(\mathcal{R}, 2 \to 1)} := \mathbb{E}\left[\sum_{t=1}^T D_{\text{KL}}\left(q(\mathcal{R}_1^{t-1} | \mathcal{R}_1^t, \mathcal{R}_1^0) || p_\theta(\mathcal{R}_1^{t-1} | \mathcal{P}_1^t, \boldsymbol{\mathcal{P}}_2^t))\right)\right], \quad (7)$$

$$\mathcal{L}^{(\mathcal{S}, 2 \to 1)} := \mathbb{E}\left[\sum_{t=1}^T D_{\text{KL}}\left(q(\mathcal{S}_1^{t-1} | \mathcal{S}_1^t, \mathcal{S}_1^0) || p_\theta(\mathcal{S}_1^{t-1} | \mathcal{P}_1^t, \boldsymbol{\mathcal{P}}_2^t))\right)\right]. \quad (8)$$

**Structure loss $\mathcal{L}^{(\mathcal{R}, 2 \to 1)}$.** It has been shown in Ho et al. (2020) that the loss function can be simplified under our parameterization by calculating KL divergence between Gaussians as weighted L2 distances between means $\epsilon_\theta$ and $\epsilon$ (see details in App. B.3):

$$\mathcal{L}^{(\mathcal{R}, 2 \to 1)} = \sum_{t=1}^T \gamma_t \mathbb{E}_{\epsilon \sim \mathcal{N}(0, I)}\left[\|\epsilon - \epsilon_\theta(\mathcal{P}_1^t, \boldsymbol{\mathcal{P}}_2^t, t)\|_2^2\right], \quad (9)$$

where the coefficients $\gamma_t$ are determined by the variances $\beta_1, ..., \beta_t$. In practice, we follow the suggestion in Ho et al. (2020) to set all weights $\gamma_t = 1$ for the simplified loss $\mathcal{L}_{\text{simple}}^{(\mathcal{R}, 2 \to 1)}$.

Since $\epsilon_\theta$ is designed to be rotation-equivariant *w.r.t.* $\mathcal{R}_1^t$, to make the loss function invariant *w.r.t.* $\mathcal{R}_1^t$, the supervision $\epsilon$ is also supposed to achieve such equivariance. So we adopt the chain-rule approach proposed in Xu et al. (2022), which decomposes the noise on pairwise distances to obtain the modified noise vector $\hat{\epsilon}$ as supervision. We refer readers to Xu et al. (2022) for more details.

**Sequence loss $\mathcal{L}^{(\mathcal{S}, 2 \to 1)}$.** Since we parameterize $p_\theta(\mathcal{S}_1^{t-1} | \mathcal{P}_1^t, \boldsymbol{\mathcal{P}}_2^t)$ with $\tilde{p}_\theta(\tilde{\mathcal{S}}_1^0 | \mathcal{P}_1^t, \boldsymbol{\mathcal{P}}_2^t)$ and $q(\mathcal{S}_1^{t-1} | \mathcal{S}_1^t, \tilde{\mathcal{S}}_1^0)$ as in (5), it can be proved that the $t$-th KL divergence term in (8) reaches zero when $\tilde{p}_\theta(\tilde{\mathcal{S}}_1^0 | \mathcal{P}_1^t, \boldsymbol{\mathcal{P}}_2^t)$ puts all mass on the ground truth $\mathcal{S}_1^0$ (see proof in App. B.4). Therefore, for the purpose of pre-training, we can simplify the KL divergence to the cross entropy loss between the correct residue type $s_{1i}^0$ and the prediction:

$$\mathcal{L}_{\text{simple}}^{(\mathcal{S}, 2 \to 1)} = \sum_{t=1}^T \sum_i \text{CE}\left(s_{1i}^0, \tilde{p}_\theta(s_{1i}^0 | \mathcal{P}_1^t, \boldsymbol{\mathcal{P}}_2^t)\right). \quad (10)$$

**Final pre-training objective.** To summarize, the ultimate training objective for our method is

$$\mathcal{L}_{\text{total}} = \frac{1}{2}\left(\mathcal{L}_{\text{simple}}^{(\mathcal{R}, 2 \to 1)} + \mathcal{L}_{\text{simple}}^{(\mathcal{S}, 2 \to 1)} + \mathcal{L}_{\text{simple}}^{(\mathcal{R}, 1 \to 2)} + \mathcal{L}_{\text{simple}}^{(\mathcal{S}, 1 \to 2)}\right), \quad (11)$$

where $\mathcal{L}_{\text{simple}}^{(\cdot, b \to a)}$ is the loss term defined by predicting trajectory $\mathcal{P}_a^{0:T}$ from trajectory $\mathcal{P}_b^{0:T}$.

## 4.4 DISCUSSION

Now we discuss the advantages of our method over previous works. The benefits of these critical designs will be empirically demonstrated by experiments in Sec. 5.3.

**Advantages of multimodal denoising.** Compared with previous works focusing on either protein sequences (Yang et al., 2022) or structures (Guo et al., 2022), in this work, we perform joint denoising on both modalities. Note that given a sequence $\mathcal{S}$ and a structure $\mathcal{R}$ that exist in the nature with high probability, the sequence-structure tuple $\mathcal{P} = (\mathcal{S}, \mathcal{R})$ may not be a valid state of this protein. Consequently, instead of modeling the marginal distribution, we are supposed to model the joint distribution of protein sequences and structures, which can be achieved by our multimodal denoising.

**Connection with diffusion models.** Diffusion models have achieved outstanding performance on image and text generation tasks (Dhariwal & Nichol, 2021; Li et al., 2022) and recently been applied on unsupervised representation learning (Abstreiter et al., 2021). The key to its success is the denoising objective at different noise levels, which has also been used for self-supervised learning in earlier works of scheduled denoising autoencoders (Geras & Sutton, 2014; Chandra & Sharma, 2014). Our method differs from these existing works by incorporating the idea of mutual prediction of two siamese diffusion trajectories so as to maximize the mutual information between them.

## 5 EXPERIMENTS

We conduct experiments on both residue and atom levels to prove the effectiveness of our method.

### 5.1 EXPERIMENTAL SETUPS

**Pre-training datasets.** Following Zhang et al. (2022), we pre-train our models with the AlphaFold protein structure database (Jumper et al., 2021; Varadi et al., 2021), including both 365K proteome-wide predicted structures and 440K Swiss-Prot (Consortium, 2021) predicted structures.

**Downstream benchmark tasks.** For downstream evaluation, we adopt the EC protein function prediction task (Gligorijević et al., 2021) and four ATOM3D tasks (Townshend et al., 2020).

1. **Enzyme Commission (EC) number prediction** task aims to predict EC numbers of proteins which describe their catalysis behavior in biochemical reactions. This task is formalized as 538 binary classification problems. We adopt the dataset splits from Gligorijević et al. (2021) and use the test split with 95% sequence identity cutoff following Zhang et al. (2022).

2. **Protein Structure Ranking (PSR)** task predicts global distance test scores of protein structure predictions submitted to the Critical Assessment of Structure Prediction (CASP) (Kryshtafovych et al., 2019) competition. This benchmark dataset is split according to the competition year.

3. **Mutation Stability Prediction (MSP)** task seeks to predict whether a mutation will increase the stability of a protein complex or not (binary classification). The benchmark dataset is split upon a 30% sequence identity cutoff among different splits.

4. **Residue Identity (RES)** task studies the structural role of an amino acid under its local environment. A model predicts the type of the center amino acid based on its surrounding atomic structure. The environments in different splits are with different protein topology classes.

**Baseline methods.** We evaluate our method on both residue- and atom-level structures. GearNet-Edge (Zhang et al., 2022) and GVP (Jing et al., 2021a) are employed as backbone models for residue and atom levels, respectively. GearNet-Edge models protein structures with different types of edges and edge-type-specific convolutions, which is further enhanced by message passing between edges. GVP constructs atom graphs and adds a vector channels for modeling equivariant features. Based on these two encoders, we compare the proposed method with previous protein structural pre-training algorithms including multiview contrastive learning (Zhang et al., 2022), denoising score matching (Guo et al., 2022) and four self-prediction methods (Zhang et al., 2022), *i.e.*, residue type, distance, angle and dihedral prediction. Details can be found in App. D.

**Training and evaluation.** For fair comparison, we pre-train our model for 50 epochs on the AlphaFold protein structure database, following Zhang et al. (2022). For downstream evaluation, we fine-tune the pre-trained models for 200 epochs on EC and 50 epochs on Atom3D tasks except RES.

Table 1: Residue-level results on EC and Atom3D tasks. We use *, † and ‡ to denote the first, second and third best performance among all models.

| | Method | EC | | PSR | | MSP | | Mean Rank |
|---|---|---|---|---|---|---|---|---|
| | | $F_{max}$ | AUPR | Global $\rho$ | Mean $\rho$ | AUROC | AUPRC | |
| | GearNet-Edge | 0.810 | 0.835 | 0.739 | 0.374 | 0.586 | 0.192 | - |
| w/ pre-training | Denoising Score Matching | 0.842 | 0.862 | 0.823 | 0.472 | 0.559 | 0.183 | 8.0 |
| | Residue Type Prediction | 0.843 | 0.870 | 0.839* | 0.497† | 0.642 | 0.229‡ | 4.3‡ |
| | Distance Prediction | 0.839 | 0.863 | 0.780 | 0.437 | 0.588 | 0.168 | 9.3 |
| | Angle Prediction | 0.853 | 0.880 | 0.794 | 0.449 | 0.632 | 0.191 | 6.3 |
| | Dihedral Prediction | 0.859‡ | 0.881‡ | 0.735 | 0.366 | 0.617 | 0.210 | 6.3 |
| | Multiview Contrast | 0.874* | 0.892* | 0.736 | 0.362 | 0.656‡ | 0.220 | 4.7 |
| | **SiamDiff** | 0.864† | 0.882† | 0.829† | 0.506* | 0.695* | 0.331* | 1.5* |
| | **SiamDiff** (*w/o* seq.&struct. diff.) | 0.802 | 0.787 | 0.726 | 0.338 | 0.626 | 0.200 | 9.7 |
| | **SiamDiff** (*w/o* seq. diff.) | 0.858 | 0.876 | 0.789 | 0.474 | 0.638 | 0.202 | 5.2 |
| | **SiamDiff** (*w/o* struct. diff.) | 0.856 | 0.873 | 0.818 | 0.481‡ | 0.660† | 0.246† | 4.0† |
| | **SiamDiff** (*w/o* MI max.) | 0.855 | 0.875 | 0.826‡ | 0.462 | 0.616 | 0.180 | 6.7 |

Table 2: Atom-level results on Atom3D tasks. Accuracy is abbreviated as Acc. We use *, † and ‡ to denote the first, second and third best performance among all models.

| | Method | PSR | | MSP | | RES | Mean Rank |
|---|---|---|---|---|---|---|---|
| | | Global $\rho$ | Mean $\rho$ | AUROC | AUPRC | Acc. | |
| | GVP | 0.809 | 0.486 | 0.652 | 0.228 | 0.550 | - |
| w/ pre-training | Denoising Score Matching | 0.849 | 0.535 | 0.625 | 0.209 | 0.558 | 7.2 |
| | Residue Type Prediction | 0.845 | 0.527 | 0.664 | 0.254 | 0.558 | 5.6 |
| | Distance Prediction | 0.825 | 0.513 | 0.621 | 0.250 | 0.504 | 8.8 |
| | Angle Prediction | 0.872* | 0.545† | 0.659 | 0.244 | 0.581† | 3.2‡ |
| | Dihedral Prediction | 0.852‡ | 0.538 | 0.692† | 0.260† | 0.585* | 2.4† |
| | Multiview Contrast | 0.848 | 0.518 | 0.640 | 0.187 | 0.335 | 9.0 |
| | **SiamDiff** | 0.854† | 0.548* | 0.694* | 0.263* | 0.563‡ | 1.6* |
| | **SiamDiff** (*w/o* seq.&struct. diff.) | 0.796 | 0.487 | 0.640 | 0.216 | 0.516 | 9.2 |
| | **SiamDiff** (*w/o* seq. diff.) | 0.850 | 0.530 | 0.670‡ | 0.255‡ | 0.472 | 5.2 |
| | **SiamDiff** (*w/o* struct. diff.) | 0.823 | 0.519 | 0.642 | 0.222 | 0.560 | 7.0 |
| | **SiamDiff** (*w/o* MI max.) | 0.850 | 0.540‡ | 0.631 | 0.217 | 0.556 | 6.2 |

Due to the large size of the RES dataset, we set the time limit as 24 hours and thus only fine-tine each model for 12 epochs. More training hyperparameters are stated in App. D.

For EC prediction, we employ $F_{max}$ and AUPR as evaluation metrics, following the original benchmark (Gligorijević et al., 2021). For PSR prediction, we utilize the global and mean Spearman's $\rho$ to assess the ranking performance. The MSP task measures the binary classification performance with AUROC and AUPRC, and the micro-averaged accuracy serves as the evaluation metric of RES. Detailed definitions of $F_{max}$, global Spearman's $\rho$ and mean Spearman's $\rho$ are stated in Appendix D.

## 5.2 EXPERIMENTAL RESULTS

Tables 1 and 2 report the results of GearNet-Edge and GVP on residue- and atom-level graphs, respectively. We do not include the results of GVP on EC and GearNet-Edge on RES tasks due to their poor performance. We analyze and discuss these results below.

**Overall, SiamDiff gains consistently superior or competitive performance on all benchmarks across residue- and atom-level structures.** On all residue- and atom-level benchmark tasks, the performance of SiamDiff is among the top three, which is not attained by previous protein structure pre-training methods. Although dihedral prediction performs well on atom-level downstream tasks, its performance on the residue-level PSR and MSP tasks is not satisfactory. The extensive experimental results demonstrate the strength of SiamDiff on boosting diverse types of structure-based protein understanding tasks, including function prediction (EC), protein model quality assessment (PSR), mutation effect prediction (MSP) and structural role understanding (RES).

**On EC, SiamDiff outperforms all baselines except Multiview Contrast.** Note that this function prediction task aims to predict their catalyzed biochemical reactions, which are mainly determined

by the structures of active/binding sites on protein surfaces. The superior performance of Multiview Contrast can be attributed to the design of its correlated views, which aims to capture the statistical relevance of these protein substructures that reflect protein functions and thus aligns representations of proteins with similar functional sites. In contrast, SiamDiff maximizes MI between siamese trajectories for modeling atom- and residue-level interactions underlying protein structural changes.

**On PSR, SiamDiff is the best in terms of mean $\rho$ and ranks the second in terms of global $\rho$.** Among all baselines, only Residue Type Prediction can achieve consistently good performance on both structure levels. It is worth noticing that both SiamDiff and residue type prediction can acquire the compatibility of protein sequence and structure. Specifically, Residue Type Prediction models the conditional distribution of sequences given structures, while SiamDiff models the joint distribution of sequences and structures. Such cross-modality modeling mechanisms benefit their performance on this task where the validity of generated structures is assessed given a protein sequence.

**On MSP, SiamDiff achieves the best performance.** This task classifies a set of similar mutant structures into two groups according to their ability to stabilize protein-protein interactions. Such classification is mostly determined by the local structure change around the mutation site, which can be well modeled by the structure diffusion process in SiamDiff. Essentially, structure diffusion guides the model to recover the original/stable structure from the perturbed/destabilized one, which is highly correlated to the objective of MSP. This can also explain why only applying the structure diffusion objective (SiamDiff *w/o* seq. diff.) still achieves the third best performance at the atom level.

**On RES, SiamDiff ranks the third place, lagging behind Angle and Dihedral Prediction.** In this task, the types of target residues are predicted given coordinates of its three backbone atoms and atoms from surrounding residues. It is known that dihedral angles on the protein backbone can determine the distribution of plausible side chain structures (Shapovalov & Dunbrack Jr, 2011) and further imply residue types. Therefore, by capturing this information explicitly or implicitly, Dihedral Prediction and Angle Prediction achieve very good results. By comparison, though SiamDiff is not specifically designed for dihedral modeling, it can still achieve competitive performance on this task.

### 5.3 ABLATION STUDY

We study different components of our method in the last blocks of Tables 1 and 2.

**Effect of multimodal diffusion.** We study three degenerated settings of multimodal diffusion, *i.e.*, *w/o* sequence and structure diffusion, *w/o* sequence diffusion and *w/o* structure diffusion. For the setting without sequence and structure diffusion, we simply perform contrastive learning between two correlated starting states from a protein. Compared to the original method, all these three settings lead to performance decay on all tasks. These results prove the necessity of both structure and sequence diffusion for SiamDiff to learn structure- and sequence-aware protein representations.

**Effect of MI maximization.** We evaluate our method under the setting without MI maximization. Specifically, we only use the original protein to derive the multimodal diffusion trajectory, and multimodal denoising is performed on this single trajectory. This setting is also clearly inferior to the original SiamDiff, since it cannot capture shared information between correlated views. Such a modeling capability is important on many tasks, *e.g.*, general protein function prediction on EC ($F_{max}$ score before and after removing this component: 0.864 *v.s.* 0.855), and it is thus an essential component of SiamDiff. More ablation studies are provided in App. C.1.

### 6 CONCLUSIONS AND FUTURE WORK

In this work, we propose the Siamese Diffusion Trajectory Prediction (SiamDiff) for protein structure pre-training. SiamDiff deems the whole diffusion trajectory of a protein as a view and seeks to maximize the mutual information between correlated trajectories. In particular, we consider the multimodal diffusion process in which both the protein folded structure and the protein sequence are gradually corrupted towards chaos, so as to capture the joint distribution of structures and sequences. Extensive experiments on diverse types of tasks and on both residue- and atom-level structures verify the consistently superior or competitive performance of SiamDiff against previous baselines.

In future works, we will enhance SiamDiff with a dedicated multimodal encoder that can better model the protein structure and sequence in a joint fashion, and we will also explore how to incorporate the large-scale protein sequence corpus into the pre-training process.

## REPRODUCIBILITY STATEMENT

For the sake of reproducibility, in App. D, we provide detailed data processing schemes, graph construction schemes, model configurations, pre-training configurations and fine-tuning configurations. All source code of this work will be released to public upon acceptance.

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

# A MORE RELATED WORKS

**Mutual Information (MI) Estimation and Maximization.** MI can measure both the linear and non-linear dependency between random variables. Some previous works (Belghazi et al., 2018; Hjelm et al., 2018) try to use neural networks to estimate the lower bound of MI, including Donsker-Varadhan representation (Donsker & Varadhan, 1983), Jensen-Shannon divergence (Fuglede & Topsoe, 2004) and Noise-Contrastive Estimation (NCE) (Gutmann & Hyvärinen, 2010; 2012). The optimization with InfoNCE loss (Oord et al., 2018) maximizes a lower bound of MI and is broadly shown to be a superior representation learning strategy (Chen et al., 2020b; Hassani & Khasahmadi, 2020; Xu et al., 2021; Liu et al., 2021; Zhang et al., 2022). In this work, we adopt the MI lower bound proposed by Liu et al. (2022) with two conditional log-likelihoods, and we formulate the learning objective by mutually denoising the multimodal diffusion processes of two correlated proteins.

# B PROOFS

## B.1 PROOF OF PROPOSITION 1

**Proof.** First, the mutual information between representations of two trajectories is defined as:

$$I(\boldsymbol{P}_1^{0:T}; \boldsymbol{P}_2^{0:T}) = \mathbb{E}_{\boldsymbol{\mathcal{P}}_1^{0:T}, \boldsymbol{\mathcal{P}}_2^{0:T} \sim p(\boldsymbol{P}_1^{0:T}, \boldsymbol{P}_2^{0:T})} \left[ \log \frac{p(\boldsymbol{\mathcal{P}}_1^{0:T}, \boldsymbol{\mathcal{P}}_2^{0:T})}{p(\boldsymbol{\mathcal{P}}_1^{0:T}) p(\boldsymbol{\mathcal{P}}_2^{0:T})} \right], \tag{12}$$

where the joint distribution is defined as $p(\boldsymbol{P}_1^{0:T}, \boldsymbol{P}_2^{0:T}) = p(\boldsymbol{P}_1^0, \boldsymbol{P}_2^0) q(\boldsymbol{P}_1^{1:T}|\boldsymbol{P}_1^0) q(\boldsymbol{P}_2^{1:T}|\boldsymbol{P}_2^0)$. Next, we can derive a lower bound with this definition:

$$
\begin{aligned}
& I(\boldsymbol{P}_1^{0:T}; \boldsymbol{P}_2^{0:T}) \\
=& \mathbb{E} \left[ \log \frac{p(\boldsymbol{\mathcal{P}}_1^{0:T}, \boldsymbol{\mathcal{P}}_2^{0:T})}{p(\boldsymbol{\mathcal{P}}_1^0) q(\boldsymbol{\mathcal{P}}_1^{1:T}|\boldsymbol{\mathcal{P}}_1^0) p(\boldsymbol{\mathcal{P}}_2^0) q(\boldsymbol{\mathcal{P}}_2^{1:T}|\boldsymbol{\mathcal{P}}_2^0)} \right] \\
\geq& \mathbb{E} \left[ \log \frac{p(\boldsymbol{\mathcal{P}}_1^{0:T}, \boldsymbol{\mathcal{P}}_2^{0:T})}{\sqrt{p(\boldsymbol{\mathcal{P}}_1^0) p(\boldsymbol{\mathcal{P}}_2^0) q(\boldsymbol{\mathcal{P}}_1^{1:T}|\boldsymbol{\mathcal{P}}_1^0) q(\boldsymbol{\mathcal{P}}_2^{1:T}|\boldsymbol{\mathcal{P}}_2^0)}} \right] \\
=& \frac{1}{2} \mathbb{E} \left[ \log \frac{p(\boldsymbol{\mathcal{P}}_1^{0:T}, \boldsymbol{\mathcal{P}}_2^{0:T})^2}{p(\boldsymbol{\mathcal{P}}_1^0) p(\boldsymbol{\mathcal{P}}_2^0) q(\boldsymbol{\mathcal{P}}_1^{1:T}|\boldsymbol{\mathcal{P}}_1^0)^2 q(\boldsymbol{\mathcal{P}}_2^{1:T}|\boldsymbol{\mathcal{P}}_2^0)^2} \right] \\
=& \frac{1}{2} \mathbb{E} \left[ \log \frac{p(\boldsymbol{\mathcal{P}}_1^{0:T}, \boldsymbol{\mathcal{P}}_2^{0:T})}{p(\boldsymbol{\mathcal{P}}_2^0) q(\boldsymbol{\mathcal{P}}_1^{1:T}|\boldsymbol{\mathcal{P}}_1^0) q(\boldsymbol{\mathcal{P}}_2^{1:T}|\boldsymbol{\mathcal{P}}_2^0)} + \log \frac{p(\boldsymbol{\mathcal{P}}_1^{0:T}, \boldsymbol{\mathcal{P}}_2^{0:T})}{p(\boldsymbol{\mathcal{P}}_1^0) q(\boldsymbol{\mathcal{P}}_1^{1:T}|\boldsymbol{\mathcal{P}}_1^0) q(\boldsymbol{\mathcal{P}}_2^{1:T}|\boldsymbol{\mathcal{P}}_2^0)} \right] \\
=& \frac{1}{2} \mathbb{E} \left[ \log \frac{p(\boldsymbol{\mathcal{P}}_1^{0:T}|\boldsymbol{\mathcal{P}}_2^{0:T})}{q(\boldsymbol{\mathcal{P}}_1^{1:T}|\boldsymbol{\mathcal{P}}_1^0)} + \log \frac{p(\boldsymbol{\mathcal{P}}_2^{0:T}|\boldsymbol{\mathcal{P}}_1^{0:T})}{q(\boldsymbol{\mathcal{P}}_2^{1:T}|\boldsymbol{\mathcal{P}}_2^0)} \right].
\end{aligned}
$$

However, since the distribution of representations are intractable to sample for optimization, we instead sample the trajectories $\mathcal{P}_1^{0:T}$ and $\mathcal{P}_2^{0:T}$ from our defined diffusion process, *i.e.*, $p(\mathcal{P}_1^{0:T}, \mathcal{P}_2^{0:T}) = p(\mathcal{P}_1^0, \mathcal{P}_2^0) q(\mathcal{P}_1^{1:T}|\mathcal{P}_1^0) q(\mathcal{P}_2^{1:T}|\mathcal{P}_2^0)$. Besides, instead of predicting representations, we use the representations from one trajectory to recover the other trajectory, which reflects more information than its representation. With these approximations, the lower bound above can be further written as:

$$\frac{1}{2} \mathbb{E} \left[ \log \frac{p(\boldsymbol{\mathcal{P}}_1^{0:T}|\boldsymbol{\mathcal{P}}_2^{0:T})}{q(\boldsymbol{\mathcal{P}}_1^{1:T}|\boldsymbol{\mathcal{P}}_1^0)} + \log \frac{p(\boldsymbol{\mathcal{P}}_2^{0:T}|\boldsymbol{\mathcal{P}}_1^{0:T})}{q(\boldsymbol{\mathcal{P}}_2^{1:T}|\boldsymbol{\mathcal{P}}_2^0)} \right] \approx \frac{1}{2} \mathbb{E}_{\mathcal{P}_1^{0:T}, \mathcal{P}_2^{0:T}} \left[ \log \frac{p(\mathcal{P}_1^{0:T}|\mathcal{P}_2^{0:T})}{q(\mathcal{P}_1^{1:T}|\mathcal{P}_1^0)} + \log \frac{p(\mathcal{P}_2^{0:T}|\mathcal{P}_1^{0:T})}{q(\mathcal{P}_2^{1:T}|\mathcal{P}_2^0)} \right]$$

We now show the first term on the right hand side can be written as the loss defined in (6). The derivation is very similar with the proof of Proposition 3 in Xu et al. (2022). We include it here for

completeness:

$$\mathbb{E}_{\mathcal{P}_1^{0:T}, \mathcal{P}_2^{0:T}} \left[ \log \frac{p(\mathcal{P}_1^{0:T} | \boldsymbol{P}_2^{0:T})}{q(\mathcal{P}_1^{1:T} | \mathcal{P}_1^0)} \right]$$

$$= \mathbb{E}_{\mathcal{P}_1^{0:T}, \mathcal{P}_2^{0:T}} \left[ \sum_{t=1}^{T} \log \frac{p(\mathcal{P}_1^{t-1} | \mathcal{P}_1^t, \boldsymbol{P}_2^{0:T})}{q(\mathcal{P}_1^t | \mathcal{P}_1^{t-1})} \right]$$

$$= \mathbb{E}_{\mathcal{P}_1^{0:T}, \mathcal{P}_2^{0:T}} \left[ \log \frac{(\mathcal{P}_1^0 | \mathcal{P}_1^1, \boldsymbol{P}_2^{0:T})}{q(\mathcal{P}_1^1 | \mathcal{P}_1^0)} + \sum_{t=2}^{T} \log \left( \frac{p(\mathcal{P}_1^{t-1} | \mathcal{P}_1^t, \boldsymbol{P}_2^{0:T})}{q(\mathcal{P}_1^{t-1} | \mathcal{P}_1^t, \mathcal{P}_1^0)} \cdot \frac{q(\mathcal{P}_1^{t-1} | \mathcal{P}_1^0)}{q(\mathcal{P}_1^t | \mathcal{P}_1^0)} \right) \right]$$

$$= \mathbb{E}_{\mathcal{P}_1^{0:T}, \mathcal{P}_2^{0:T}} \left[ -\log q(\mathcal{P}_1^T | \mathcal{P}_1^0) + \log p(\mathcal{P}_1^0 | \mathcal{P}_1^1, \boldsymbol{P}_2^{0:T}) + \sum_{t=2}^{T} \log \frac{p(\mathcal{P}_1^{t-1} | \mathcal{P}_1^t, \boldsymbol{P}_2^{0:T})}{q(\mathcal{P}_1^{t-1} | \mathcal{P}_1^t, \mathcal{P}_1^0)} \right]$$

$$= -\mathbb{E}_{\mathcal{P}_1^{0:T}, \mathcal{P}_2^{0:T}} \left[ \sum_{t=1}^{T} D_{\mathrm{KL}} \left( q(\mathcal{P}_1^{t-1} | \mathcal{P}_1^t, \mathcal{P}_1^0) || p(\mathcal{P}_1^{t-1} | \mathcal{P}_1^t, \boldsymbol{P}_2^{0:T}) \right) \right] + C^{(2 \to 1)}$$

$$= -\mathcal{L}^{(2 \to 1)} + C^{(2 \to 1)},$$

where we merge the term $p(\mathcal{P}_1^0 | \mathcal{P}_1^1, \boldsymbol{P}_2^{0:T})$ into the sum of KL divergences for brevity and use $C^{(2 \to 1)}$ to denote the constant independent of our encoder. Note that the counterpart can be derived in the same way. Adding these two terms together finishes the proof of Proposition 1. □

## B.2 PROOF OF PRE-TRAINING LOSS DECOMPOSITION

We restate the proposition of pre-training loss decomposition rigorously as below.

**Proposition 2** *Given the assumptions 1) the separation of the diffusion process on protein structures and sequences*

$$q(\mathcal{P}_a^t | \mathcal{P}_a^{t-1}) = q(\mathcal{R}_a^t | \mathcal{R}_a^{t-1}) \cdot q(\mathcal{S}_a^t | \mathcal{S}_a^{t-1}), \tag{13}$$

*and 2) the conditional independence of the generation process*

$$p_\theta(\mathcal{P}_a^{t-1} | \mathcal{P}_a^t, \boldsymbol{P}_b^t) = p_\theta(\mathcal{R}_a^{t-1} | \mathcal{P}_a^t, \boldsymbol{P}_b^t) \cdot p_\theta(\mathcal{S}_a^{t-1} | \mathcal{P}_a^t, \boldsymbol{P}_b^t), \tag{14}$$

*it can be proved that*

$$\mathcal{L}^{(b \to a)} = \mathcal{L}^{(\mathcal{R}, b \to a)} + \mathcal{L}^{(\mathcal{S}, b \to a)}, \tag{15}$$

*where the three loss terms are defined as*

$$\mathcal{L}^{(b \to a)} := \mathbb{E} \left[ \sum_{t=1}^{T} D_{KL} \left( q(\mathcal{P}_a^{t-1} | \mathcal{P}_a^t, \mathcal{P}_a^0) || p_\theta(\mathcal{P}_a^{t-1} | \mathcal{P}_a^t, \boldsymbol{P}_b^t) \right) \right],$$

$$\mathcal{L}^{(\mathcal{R}, b \to a)} := \mathbb{E} \left[ \sum_{t=1}^{T} D_{KL} \left( q(\mathcal{R}_a^{t-1} | \mathcal{R}_a^t, \mathcal{R}_a^0) || p_\theta(\mathcal{R}_a^{t-1} | \mathcal{P}_a^t, \boldsymbol{P}_b^t) \right) \right],$$

$$\mathcal{L}^{(\mathcal{S}, b \to a)} := \mathbb{E} \left[ \sum_{t=1}^{T} D_{KL} \left( q(\mathcal{S}_a^{t-1} | \mathcal{S}_a^t, \mathcal{S}_a^0) || p_\theta(\mathcal{S}_a^{t-1} | \mathcal{P}_a^t, \boldsymbol{P}_b^t) \right) \right],$$

*with $b \to a$ referring to the term from trajectory $\mathcal{P}_b^{0:T}$ to $\mathcal{P}_a^{0:T}$.*

**Proof.** Let $\mathcal{L}_t^{(\cdot)}$ to denote the t-th KL divergence term in $\mathcal{L}^{(\cdot)}$. Then, we have

$$\mathcal{L}_t^{(b \to a)} = D_{\mathrm{KL}} \left( q(\mathcal{P}_a^{t-1} | \mathcal{P}_a^t, \mathcal{P}_a^0) || p_\theta(\mathcal{P}_a^{t-1} | \mathcal{P}_a^t, \boldsymbol{P}_b^{0:T}) \right)$$

$$= D_{\mathrm{KL}} \left( \left[ q(\mathcal{R}_a^{t-1} | \mathcal{R}_a^t, \mathcal{R}_a^0) q(\mathcal{S}_a^{t-1} | \mathcal{S}_a^t, \mathcal{S}_a^0) \right] || \left[ p_\theta(\mathcal{R}_a^{t-1} | \mathcal{P}_a^t, \boldsymbol{P}_b^{0:T}) p_\theta(\mathcal{S}_a^{t-1} | \mathcal{P}_a^t, \boldsymbol{P}_b^{0:T}) \right] \right)$$

$$= D_{\mathrm{KL}} \left( q(\mathcal{R}_a^{t-1} | \mathcal{R}_a^t, \mathcal{R}_a^0) || p_\theta(\mathcal{R}_a^{t-1} | \mathcal{P}_a^t, \boldsymbol{P}_b^{0:T}) \right) + D_{\mathrm{KL}} \left( q(\mathcal{S}_a^{t-1} | \mathcal{S}_a^t, \mathcal{S}_a^0) || p_\theta(\mathcal{S}_a^{t-1} | \mathcal{P}_a^t, \boldsymbol{P}_b^{0:T}) \right)$$

$$= \mathcal{L}_t^{(\mathcal{R}, b \to a)} + \mathcal{L}_t^{(\mathcal{S}, b \to a)},$$

where we use the assumptions (13) and (14) in the second equality. The third equality is due to the additive property of the KL divergence for independent distributions. Adding $T$ KL divergence terms together will lead to (15). □

### B.3 PROOF OF SIMPLIFIED STRUCTURE LOSS

For completeness, we show how to derive the simplified structure loss. The proof is directly adapted from (Xu et al., 2022).

**Proposition 3** *Given the definition of the forward process*

$$q(\mathcal{R}_a^t|\mathcal{R}_a^{t-1}) = \mathcal{N}(\mathcal{R}_a^t; \sqrt{1-\beta_t}\mathcal{R}_a^{t-1}, \beta_t I), \tag{16}$$

*and the reverse process*

$$p_\theta(\mathcal{R}_a^{t-1}|\mathcal{P}_a^t, \boldsymbol{P}_b^t) = \mathcal{N}(\mathcal{R}^{t-1}; \mu_\theta(\mathcal{P}_a^t, \boldsymbol{P}_b^t, t), \sigma_t^2 I), \tag{17}$$

$$\mu_\theta(\mathcal{P}_a^t, \boldsymbol{P}_b^t, t) = \frac{1}{\sqrt{\alpha_t}}\left(\mathcal{R}_a^t - \frac{\beta_t}{\sqrt{1-\bar{\alpha}_t}}\epsilon_\theta(\mathcal{P}_a^t, \boldsymbol{P}_b^t, t)\right), \tag{18}$$

*the structure loss function*

$$\mathcal{L}^{(\mathcal{R},b\to a)} := \mathbb{E}\left[\sum_{t=1}^T D_{KL}\left(q(\mathcal{R}_a^{t-1}|\mathcal{R}_a^t, \mathcal{R}_a^0)||p_\theta(\mathcal{R}_a^{t-1}|\mathcal{P}_a^t, \boldsymbol{P}_b^t))\right], \tag{19}$$

*can be simplified to*

$$\mathcal{L}^{(\mathcal{R},b\to a)} = \sum_{t=1}^T \gamma_t \mathbb{E}_{\epsilon\sim\mathcal{N}(0,I)}\left[\|\epsilon - \epsilon_\theta(\mathcal{P}_a^t, \boldsymbol{P}_b^t, t)\|_2^2\right], \tag{20}$$

*where $\gamma_t = \frac{\beta_t}{2\alpha_t(1-\bar{\alpha}_{t-1})}$ with $\alpha_t = 1 - \beta_t$, $\bar{\alpha}_t = \prod_{s=1}^t \alpha_s$ and $b \to a$ is either $2 \to 1$ or $1 \to 2$.*

**Proof.** First, we prove $q(\mathcal{R}_a^t|\mathcal{R}_a^0) = \mathcal{N}(\mathcal{R}_a^t; \sqrt{\bar{\alpha}_t}\mathcal{R}_a^0, (1-\bar{\alpha}_t)I)$. Let $\epsilon_i$ be the standard Gaussian random variable at time step $i$. Then, we have

$$\begin{aligned}
\mathcal{R}_a^t &= \sqrt{\alpha_t}\mathcal{R}_a^{t-1} + \sqrt{\beta_t}\epsilon_t \\
&= \sqrt{\alpha_{t-1}\alpha_t}\mathcal{R}_a^{t-2} + \sqrt{\alpha_{t-1}\beta_{t-1}}\epsilon_{t-1} + \sqrt{\beta_t}\epsilon_t \\
&= \cdots \\
&= \sqrt{\bar{\alpha}_t}\mathcal{R}_a^0 + \sqrt{\alpha_t\alpha_{t-1}...\alpha_2\beta_1}\epsilon_1 + \cdots + \sqrt{\alpha_{t-1}\beta_{t-1}}\epsilon_{t-1} + \sqrt{\beta_t}\epsilon_t,
\end{aligned}$$

which suggests that the mean of $\mathcal{R}_a^t$ is $\sqrt{\bar{\alpha}_t}\mathcal{R}_a^0$ and the variance matrix is $(\alpha_t\alpha_{t-1}...\alpha_2\beta_1 + \cdots + \alpha_{t-1}\beta_{t-1} + \beta_t)I = (1-\bar{\alpha})I$.

Next, we derive the posterior distribution as:

$$\begin{aligned}
q(\mathcal{R}_a^{t-1}|\mathcal{R}_a^t, \mathcal{R}_a^0) &= \frac{q(\mathcal{R}_a^t|\mathcal{R}_a^{t-1})q(\mathcal{R}_a^{t-1}|\mathcal{R}_a^0)}{q(\mathcal{R}_a^t|\mathcal{R}_a^0)} \\
&= \frac{\mathcal{N}(\mathcal{R}_a^t; \sqrt{\alpha_t}\mathcal{R}_a^{t-1}, \beta_t I) \cdot \mathcal{N}(\mathcal{R}_a^{t-1}; \sqrt{\bar{\alpha}_{t-1}}\mathcal{R}_a^0, (1-\bar{\alpha}_{t-1})I)}{\mathcal{N}(\mathcal{R}_a^t; \sqrt{\bar{\alpha}_t}\mathcal{R}_a^0, (1-\bar{\alpha}_t)I)} \\
&= \mathcal{N}(\mathcal{R}^{t-1}; \frac{\sqrt{\bar{\alpha}_{t-1}}\beta_t}{1-\bar{\alpha}_t}\mathcal{R}_a^0 + \frac{\sqrt{\alpha_t}(1-\bar{\alpha}_{t-1})}{1-\bar{\alpha}_t}\mathcal{R}_a^t, \frac{1-\bar{\alpha}_{t-1}}{1-\bar{\alpha}_t}\beta_t I).
\end{aligned}$$

Let $\tilde{\beta}_t = \frac{1-\bar{\alpha}_{t-1}}{1-\bar{\alpha}_t}\beta_t$, then the $t$-th KL divergence term can be written as:

$$\begin{aligned}
&D_{\text{KL}}\left(q(\mathcal{R}_a^{t-1}|\mathcal{R}_a^t, \mathcal{R}_a^0)||p_\theta(\mathcal{R}_a^{t-1}|\mathcal{P}_a^t, \boldsymbol{P}_b^t)\right) \\
&= \frac{1}{2\tilde{\beta}_t}\left\|\frac{\sqrt{\bar{\alpha}_{t-1}}\beta_t}{1-\bar{\alpha}_t}\mathcal{R}_a^0 + \frac{\sqrt{\alpha_t}(1-\bar{\alpha}_{t-1})}{1-\bar{\alpha}_t}\mathcal{R}_a^t - \frac{1}{\sqrt{\alpha_t}}\left(\mathcal{R}_a^t - \frac{\beta_t}{\sqrt{1-\bar{\alpha}_t}}\epsilon_\theta(\mathcal{P}_a^t, \boldsymbol{P}_b^t, t)\right)\right\|^2 \\
&= \frac{1}{2\tilde{\beta}_t}\mathbb{E}_\epsilon\left\|\frac{\sqrt{\bar{\alpha}_{t-1}}\beta_t}{1-\bar{\alpha}_t}\cdot\frac{\mathcal{R}_a^t - \sqrt{1-\bar{\alpha}_t}\epsilon}{\sqrt{\bar{\alpha}_t}} + \frac{\sqrt{\alpha_t}(1-\bar{\alpha}_{t-1})}{1-\bar{\alpha}_t}\mathcal{R}_a^t - \frac{1}{\sqrt{\alpha_t}}\left(\mathcal{R}_a^t - \frac{\beta_t}{\sqrt{1-\bar{\alpha}_t}}\epsilon_\theta(\mathcal{P}_a^t, \boldsymbol{P}_b^t, t)\right)\right\|^2 \\
&= \frac{1}{2\tilde{\beta}_t}\cdot\frac{\beta_t^2}{\alpha_t(1-\bar{\alpha}_t)}\mathbb{E}_\epsilon\left\|\epsilon - \epsilon_\theta(\mathcal{P}_a^t, \boldsymbol{P}_b^t, t)\right\| \\
&= \gamma_t\mathbb{E}_\epsilon\left[\|\epsilon - \epsilon_\theta(\mathcal{P}_a^t, \boldsymbol{P}_b^t, t)\|_2^2\right],
\end{aligned}$$

which completes the proof. $\square$

### B.4 Proof of Simplified Sequence Loss

Now we show the equivalence of optimizing sequence loss $\mathcal{L}^{(\mathcal{S}, b \to a)}$ and the masked residue type prediction problem on $\mathcal{S}_a^0$.

**Proposition 4** *Given the definition of reverse process on protein sequences*

$$p_\theta(\mathcal{S}_a^{t-1}|\mathcal{P}_a^t, \boldsymbol{\mathcal{P}}_b^t) \propto \sum_{\tilde{\mathcal{S}}_a^0} q(\mathcal{S}_a^{t-1}|\mathcal{S}_a^t, \tilde{\mathcal{S}}_a^0) \cdot \tilde{p}_\theta(\tilde{\mathcal{S}}_a^0|\mathcal{P}_a^t, \boldsymbol{\mathcal{P}}_b^t), \tag{21}$$

*the sequence loss $\mathcal{L}^{(\mathcal{S}, b \to a)}$ reaches zero when $\tilde{p}_\theta(\tilde{\mathcal{S}}_a^0|\mathcal{P}_a^t, \boldsymbol{\mathcal{P}}_b^t)$ puts all mass on the ground truth $\mathcal{S}_a^0$.*

**Proof.** The loss function can be written as:

$$\mathcal{L}^{(\mathcal{S}, b \to a)} := \mathbb{E}\left[\sum_{t=1}^T D_{\mathrm{KL}}\left(q(\mathcal{S}_a^{t-1}|\mathcal{S}_a^t, \mathcal{S}_a^0)||p_\theta(\mathcal{S}_a^{t-1}|\mathcal{P}_a^t, \boldsymbol{\mathcal{P}}_b^t)\right)\right]$$

$$= \mathbb{E}\left[\sum_{t=1}^T D_{\mathrm{KL}}\left(q(\mathcal{S}_a^{t-1}|\mathcal{S}_a^t, \mathcal{S}_a^0)\middle\|\frac{\sum_{\tilde{\mathcal{S}}_a^0} q(\mathcal{S}_a^{t-1}|\mathcal{S}_a^t, \tilde{\mathcal{S}}_a^0) \cdot \tilde{p}_\theta(\tilde{\mathcal{S}}_a^0|\mathcal{P}_1^t, \boldsymbol{\mathcal{P}}_2^t)}{Z}\right)\right],$$

where $Z$ is the normalization constant. Hence, when $\tilde{p}_\theta(\tilde{\mathcal{S}}_a^0|\mathcal{P}_a^t, \boldsymbol{\mathcal{P}}_b^t)$ puts all mass on the ground truth $\mathcal{S}_a^0$, the distribution $p_\theta(\mathcal{S}_a^{t-1}|\mathcal{P}_a^t, \boldsymbol{\mathcal{P}}_b^t)$ will be identical with $q(\mathcal{S}_a^{t-1}|\mathcal{S}_a^t, \mathcal{S}_a^0)$, which makes the KL divergence become zero. $\square$

## C More Experimental Results

### C.1 Ablation Study

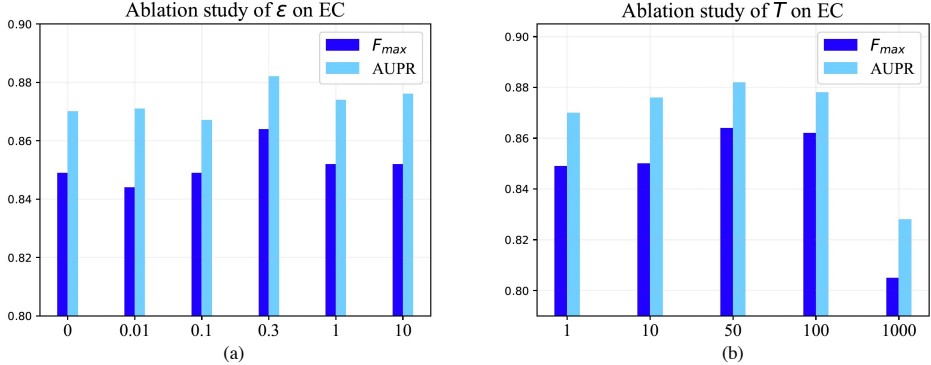

Figure 2: Ablation study of (a) structure perturbation scale $\epsilon$ and (b) time step number $T$ on EC.

**Effect of structure perturbation scale $\epsilon$.** In Fig. 2(a), we present the performance of SiamDiff on EC under different structure perturbation scales. We can observe the superiority of $\epsilon = 0.3$ over other settings on both evaluation metrics. This result illustrates that moderately large structure perturbations favor SiamDiff's effectiveness. Under such condition, the correlated views can distinguish from each other while share sufficient common information, leading to the moderate difficulty of the SiamDiff pre-training task.

**Effect of time step number $T$.** Fig. 2(b) shows SiamDiff's performance on EC by using different time step numbers. It can be observed that too few time steps (*i.e.*, 1 or 10 time steps) and too many time steps (*i.e.*, 1000 steps) both lead to the inferior performance. In these situations, the task of each denoising step is either too hard (with too few time steps) or too simple (with too many time steps). Therefore, it is suggested to use moderately many time steps (*e.g.*, 50 or 100 time steps) in the SiamDiff pre-training method.

## D Experimental Details

In this section, we introduce the details of our experiments. All these methods are developed based on PyTorch and TorchDrug (Zhu et al., 2022).

**Graph construction.** For atom graphs, we connect atoms with Euclidean distance lower than a distance threshold. For PSR and MSP tasks, we remove all hydrogen atoms following Jing et al. (2021b). For residue graphs, we discard all non-alpha-carbon atoms and add three different types of directed edges: sequential edges, radius edges and K-nearest neighbor edges. For sequential edges, two atoms are connected if their sequential distance is below a threshold and these edges are divided into different types according to these distances. For two kinds of spatial edges, we connect atoms according to Euclidean distance and k-nearest neighbors. We further apply a long range interaction filter that removes edges with low sequential distances.

**Backbone models.** We directly borrow the best hyperparameters reported in the original paper of GearNet-Edge (Zhang et al., 2022). We adopt the same configuration of relational graph construction, *i.e.*, the sequential distance threshold $d_{\text{seq}} = 3$, the radius $d_{\text{radius}} = 10.0$Å, the number of neighbors $k = 10$ and the long range interaction cutoff $d_{\text{long}} = 5$. We use one-hot features of residue types as node features and concatenate (1) one-hot features of end nodes, (2) one-hot features of edge types, (3) sequential distance, (4) spatial distance as edge features. Then we use 6 message passing layers with 512 hidden dimensions and ReLU as the activation function. For edge message passing, the edge types on the line graph are determined by the discretized angles. The hidden representations in each layer of GearNet will be concatenated for the final prediction. Since only alpha carbon atoms are kept in the graph, their representations are used for both atom and residue representations.

For GVP, the original design only includes atom types as node features, which makes pre-training tasks with residue type prediction very difficult to learn. To address this issue, we slightly modify its architecture to add add the embedding of atom and corresponding residue types as atom features. Then, the default configurations in Jing et al. (2021b) are adopted. We construct an atom graph for each protein by drawing edges between atoms closer than $4.5$Å. Each edge is featured with a 16-dimensional Gaussian RBF encoding of its Euclidean distance. We use five GVP layers and hidden representations with 16 vector and 100 scalar channels and use ReLU and identity for scalar and vector activation functions, respectively. The dropout rate is set as 0.1. The final atom representations are followed by a mean pooling layer to obtain residue and protein representations.

**Pre-training methods.** Here we briefly introduce the considered baselines. Multiview Contrast aims to maximize the mutual information between correlated views, which are extracted by randomly chosen augmentation functions to capture protein sub-structures. Residue type, distance, angle and dihedral prediction masks single residues, single edges, edge pairs and edge triplets, respectively, and then predict the corresponding properties. Denoising score matching performs denoising on noised pairwise distance matrices based on the learnt representations.

For all baselines in (Zhang et al., 2022), we adopt the original configurations. For Multiview Contrast, we use subsequence cropping that randomly extracts protein subsequences with no more than 50 residues and space cropping that takes all residues within a $15$Å Euclidean ball with a random center residue. Then, either an identity function or a random edge masking function with mask rate equal to 0.15 is applied for constructing views. The temperature $\tau$ in the InfoNCE loss function is set as 0.07. We set the number of sampled items in each protein as 256 for Distance Prediction and as 512 for Angle and Dihedral Prediction. The mask rate for Residue Type Prediction is set as 0.15. When masking a residue on atom graphs, we discard all non-backbone atoms and set the residue features as zero. Since the backbone models and tasks in our paper are quite different with those in Guo et al. (2022), we re-implement the method on our codebase. We consider 50 different noise levels log-linearly ranging from 0.01 to 10.0.

For our method, we set the variance of structure perturbation noises $\epsilon$ as 0.3 when constructing the other view. For structure diffusion, we use a sigmoid schedule for variances $\beta_t$ with the lowest variance $\beta_1 = 1e-3$ and the highest variance $\beta_T = 0.1$. For sequence diffusion, we simply set the cumulative transition probability to [MASK] over time steps as a linear interpolation between minimum mask rate 0.15 and maximum mask rate 1.0. The number of diffusion steps is set as 1000.

All other optimization configurations for these pre-training methods are reported in Table 3. All methods are pre-trained on four Tesla A100 GPUs and Table 3 reports the batch sizes on each GPU.

**Fine-tuning on downstream tasks.** For all models on all downstream tasks, we apply the a three-layer MLP head for prediction, the hidden dimension of which is set to the dimension of model

Table 3: Optimization configurations for pre-training methods. Here max length denotes the maximum number of residues kept in each protein and lr stands for learning rate.

| Method | Max length | | Batch size | | Optimizer | lr |
|---|---|---|---|---|---|---|
| | residue | atom | residue | atom | | |
| Residue Type Prediction | 100 | 100 | 96 | 64 | Adam | 1e-3 |
| Distance Prediction | 100 | 100 | 128 | 64 | Adam | 1e-3 |
| Angle Prediction | 100 | 100 | 96 | 64 | Adam | 1e-3 |
| Dihedral Prediction | 100 | 100 | 96 | 64 | Adam | 1e-3 |
| Multiview Contrast | - | - | 96 | 64 | Adam | 1e-3 |
| Denoising Score Matching | 200 | 200 | 12 | 12 | Adam | 1e-4 |
| **SiamDiff** | 150 | 100 | 32 | 32 | Adam | 1e-4 |

outputs. The batch sizes for each model are chosen according the memory limit. For all residue-level tasks, we fine-tune the GearNet-Edge on 4 GPUs (A100 for EC/GO and V100 for PSR/MSP) with the batch size on a single one as 2. For all atom-level tasks except RES, we set the batch size as 2 and fine-tune the model on a single V100 GPU. For RES, we fine-tune the pre-trained GVP on 4 A100 GPUs with the batch size on a single one as 32.

**Evaluation metrics.** We clarify the definitions of $F_{max}$ (used in EC), global Spearman's $\rho$ (used in PSR) and mean Spearman's $\rho$ (used in PSR) as below:

- **$F_{max}$** denotes the protein-centric maximum F-score. It first computes the precision and recall for each protein at a decision threshold $t \in [0, 1]$:

$$\text{precision}_i(t) = \frac{\sum_f \mathbb{1}[f \in P_i(t) \cap T_i]}{\sum_f \mathbb{1}[f \in P_i(t)]}, \quad \text{recall}_i(t) = \frac{\sum_f \mathbb{1}[f \in P_i(t) \cap T_i]}{\sum_f \mathbb{1}[f \in T_i]}, \quad (22)$$

where $f$ denotes a functional term in the ontology, $T_i$ is the set of experimentally determined functions for protein $i$, $P_i(t)$ is the set of predicted functions for protein $i$ whose scores are greater or equal to $t$, and $\mathbb{1}[\cdot]$ represents the indicator function. After that, the precision and recall are averaged over all proteins:

$$\text{precision}(t) = \frac{1}{M(t)} \sum_i \text{precision}_i(t), \quad \text{recall}(t) = \frac{1}{N} \sum_i \text{recall}_i(t), \quad (23)$$

where $N$ denotes the total number of proteins, and $M(t)$ denotes the number of proteins which contain at least one prediction above the threshold $t$, *i.e.*, $|P_i(t)| > 0$.

Based on these two metrics, the $F_{max}$ score is defined as the maximum value of F-measure over all thresholds:

$$F_{max} = \max_t \left\{ \frac{2 \cdot \text{precision}(t) \cdot \text{recall}(t)}{\text{precision}(t) + \text{recall}(t)} \right\}. \quad (24)$$

- **Global Spearman's $\rho$ for PSR** measures the correlation between the predicted global distance test (GDT_TS) score and the ground truth. It computes the Spearman's $\rho$ between the prediction and the ground truth over all test proteins without considering the different biopolymers that these proteins lie in.

- **Mean Spearman's $\rho$ for PSR** also measures the correlation between GDT_TS predictions and the ground truth. However, it first splits all test proteins into multiple groups based on their corresponding biopolymers, then computes the Spearman's $\rho$ within each group, and finally reports the mean Spearman's $\rho$ over all groups.

