# OpenReview forum: "Pre-training Protein Structure Encoder via Siamese Diffusion Trajectory Prediction"
_ICLR.cc/2023/Conference — Submitted to ICLR 2023_

### Official Review · Reviewer_UBXw · 2022-10-14

**Confidence:** 1
**Correctness:** 3
**Technical Novelty And Significance:** 2
**Empirical Novelty And Significance:** 2
**Recommendation:** 5

**Clarity, Quality, Novelty And Reproducibility:**

Clarity: good.

Quality: good.

Novelty: Low.

Reproducibility: unknown. No example code was provided.

**Strength And Weaknesses:**

Strength:
Using diffusion model as a pretrain model seems to be a new application.

Weakness:
1. I am not an expert of the application area this paper considers. But simply examing the experiment, it seems the improvement is not large. There is no repeated experiment and it's unknown whether the performance is significant.

2. The overall technical novelty is low. It follows a standard design of diffusion model and the contribution seems restricted to diffusion as a pretrain model?

**Summary Of The Paper:**

This paper proposed to use diffusion model as a pretrain model for several downstream tasks.

It decomposes the structure and uses a DDPM like diffusion for continuous variable and a multimodel diffusion for categorical variable. Experiment shows that the diffusion model serves as a good pretrain model.

**Summary Of The Review:**

See above.

My evaluation is a educated guess.

---

> ### Author Response · Authors · 2022-11-15
> **Author Feedbacks to Reviewer UBXw**
>
> Thanks for your insightful comments and golden suggestions! We respond to your concerns as below:
>
> >**Q1: There is no repeated experiment and it's unknown whether the performance gain is significant.**
>
> Thanks for reminding us! We admit that reporting standard deviation is important. However, it is infeasible for us to finish these numerous experiments during rebuttal ($n \times 12 \times 6$ experiments in total, where $n$ is the number of repeated experiments). We’ll continue to work on this and report the numbers in the final version.
>
> >**Q2: The overall technical novelty is low. It follows a standard design of diffusion model and the contribution seems restricted to diffusion as a pretrain model.**
>
> We would like clarify that the novelty of our method mainly lies in the following two aspects:
>
> 1. **Combining the advantages of diffusion modeling and mutual information (MI) maximization on protein representation learning:** For protein representation learning, diffusion modeling and MI maximization show their strengths in different aspects: (1) **Diffusion modeling excels at modeling the dependencies within a single protein**, in which the atom and residue interactions along protein formation can be captured by the model; (2) By comparison, **MI maximization based methods are good at modeling the global dependencies among different proteins**, where the representations of correlated proteins are constrained to be nearby in the latent space. By combining these orthogonal objectives of representation learning, **the learned model is expected to produce both (1) residue/atom interaction-aware representations and (2) structural similarity-aware representations.** Such merits cannot be attained by previous methods [a,b] performing only MI maximization or a method performing only diffusion modeling.
>
> **This novel contribution is well supported by our ablation studies.** In Table 1 and 2, by comparing the performance of SiamDiff with its ablation variants without either diffusion modeling (the first line of last block) or MI maximization (the last line of last block), we can observe clear performance gains of SiamDiff which combines the two learning mechanisms.
>
> 2. **Joint modeling of protein structures and sequences by a novel multimodal diffusion strategy:** Compared to the conventional diffusion modeling that diffuses either continuous or discrete random variables, **the proposed multimodal diffusion trajectory simultaneously diffuses the continuous protein structure and the discrete protein sequence.** Such a scheme well fits the protein representation learning problem where the structure and the sequence of a protein can reflect different protein properties and are thus complementary to each other.
>
> **This novel design is well supported by our ablation studies.** In Table 1 and 2, by comparing the performance of SiamDiff with its ablation variants without either sequence diffusion (the second line of last block) or structure diffusion (the third line of last block), we can observe non-trivial performance gains of SiamDiff which combines both structure and sequence diffusion.
>
> **Based on the two technical contributions above, we argue for the decent novelty of our proposed method.**
>
> &emsp;
>
> [a] Protein representation learning by geometric structure pretraining. Zhang et al., arXiv:2203.06125 (2022).
>
> [b] Contrastive representation learning for 3d protein structures. Hermosilla  et al., arXiv:2205.15675 (2022).

---

### Official Review · Reviewer_jQVb · 2022-10-23

**Confidence:** 4
**Correctness:** 4
**Technical Novelty And Significance:** 3
**Empirical Novelty And Significance:** 3
**Recommendation:** 8

**Clarity, Quality, Novelty And Reproducibility:**

The paper is overall well-written and the language is concise. However, as I mentioned before, I feel that the paper is not that clear. I had to go through several readings of the manuscript to fully comprehend the main ideas in the paper. Some better intuition and improving of the structure could be greatly beneficial to this paper.

I believe this is a high-quality paper. The mathematical framework introduced is sound, the experiments are extensive and thorough, and the empirical results obtained are strong.

This method is a novel application of existing methods, however, the real novelty lies in the ideas applied to learn the protein representations. There are some clear distinctions from previous works, like the multimodal denoising and the Siamese paired views of proteins.

As for reproducibility, the authors mention that the code will be made available in case the paper gets accepted. The representation learning is well-defined in the paper. However, the models used for downstream tasks, based on the representations, are not that clear. The information can be found in the supplementary material, but I believe that a couple of sentences providing more details about these models should be included in the main paper. In short, releasing the code for both the training of SiamDiff, as well as the pipeline for downstream tasks is critical for the reproducibility of this work.


**Strength And Weaknesses:**

The authors introduce some interesting and novel ideas in this paper. They show that their Siamese multimodal diffusion process can obtain reasonable protein representations that are useful for several different downstream tasks. Modelling the joint distribution of protein sequences and structures through multimodal denoising allows them to capture SiamDiff to capture the atom and residue-level interaction. Another interesting point is their idea of maximising the MI between two Siamese diffusion trajectories, one based on the native view of the protein, and the other view being generated by a structure perturbation.

The mathematical framework presented is sound. The authors also present the proofs to every proposition they make. They also mention that the full code will be available for reproducibility.

The experiments shown in the paper are very thorough, where all the tasks have standardised experimental settings in well-established benchmarks. The empirical results are strong enough to prove the effectiveness of the protein representations obtained by SiamDiff.

I feel that the paper is lacking clarity. While I understand that there are a lot of concepts to fit in the manuscript, I believe that this paper could benefit from high-level intuition behind the main ideas of the paper. The notation feels overloaded and it is hard to follow.


**Summary Of The Paper:**

In this paper, the authors propose SiamDiff, a method that employs a multimodal diffusion process to simulate the structure-sequence co-diffusion trajectory. SiamDiff maximises the Mutual Information (MI) between paired correlated views of proteins, where one is the native protein and the other is obtained with a structure perturbation. Under this learning framework, the authors argue that the model can acquire the atom- and residue-level interactions underlying protein structural changes, as well as the residue type dependencies. Extensive experiments on diverse and standardised downstream tasks are shown to demonstrate the value of the learned protein representations.

**Summary Of The Review:**

My recommendation is an accept. I believe that this is a good paper introducing some novel ideas that are very relevant to protein representation learning. The evaluation of the learned representations in downstream tasks was made following well-established benchmarks. Although, the performance achieved is not the highest for every task, the general value of these representations is evident from the empirical results.
I do expect the authors to address the issues raised in this review, mainly about clarity.

---

> ### Author Response · Authors · 2022-11-15
> **Author Feedbacks to Reviewer jQVb**
>
> Thanks for your appreciation of our work! We respond to your concerns as below:
>
> >**Q1: I feel that the paper is lacking clarity. I believe that this paper could benefit from high-level intuition behind the main ideas of the paper.**
>
> This suggestion is great! To enhance the readability of this paper and make it more accessible to the audience, we have done following revisions:
>
> 1. In the revised Abstract and Introduction, **we give a more comprehensible definition of “correlated views” as “different descriptions of the same protein”**, so as to avoid the confusion of the audience that are not quite familiar with contrast-based protein representation learning.
>
> 2. To better motivate the audience from the general community and make the main idea of our method more accessible, in the revised paper, **we add two paragraphs at the beginning of Section 4.1 to better clarify (1) two main limitations of existing protein structure pre-training methods based on mutual information maximization, and (2) how the proposed multimodal diffusion trajectory can help to tackle these limitations.** We hope these contents can make our starting point and main idea more accessible.
>
> Please check out these revisions! We welcome any feedback to further enhance the readability of this paper.
>
> >**Q2: The models used for downstream tasks, based on the representations, are not that clear. A couple of sentences providing more details about these models should be included in the main paper.**
>
> Thanks for the suggestions! We have added detailed descriptions about the backbone models and pre-training methods in Sec. 5.1 and App. D.

---

### Official Review · Reviewer_kp2m · 2022-10-25

**Confidence:** 3
**Clarity, Quality, Novelty And Reproducibility:** It is quite clear however the author …
**Correctness:** 2
**Technical Novelty And Significance:** 2
**Empirical Novelty And Significance:** 2
**Recommendation:** 5

**Strength And Weaknesses:**

Strength:
The proposed method tried to achieve better protein residual presentation by introducing contrast learning and maximize mutual information.
The proposed method showed marginal performance increase compared with existing methods.

Weakness:
As far as I regard, the proposed method is not quite intuitive. Diffusion model is already a method that apply graduate noise. Such that, introducing another level of random noise for contrast learning is not intuitive to me. The marginal performance increase also suggest some conflict between diffusion model and the contrast learning. I was wondering is there some theoretical analysis between diffusion model and the introduced random noise?
I also do not quite agree with the claim that added random noise could reflect the conformational fluctuation. Simply add the noise on the structure is not equivalent to random permutation. Sometimes, a random permutation could drastically change the protein structure. Simply add random noise, could never reflect such conformational change.



**Summary Of The Paper:**

This paper applied diffusion model to learn the protein structure from protein sequence. Compared with previous method, they introduced contrast learning and utilized mutual information. The experiment revealed that the proposed method showed marginal performance increase.

**Summary Of The Review:**

The paper focus on the residual level prediction accuracy. It utilize diffusion model with added mutual information training. However, there lack some clear linkage between diffusion model and synthetic mutual information, which makes the method not intuitive. Moreover, the added noise is quite straightforward, which is not optimal to reflect conformation change.

---

> ### Author Response · Authors · 2022-11-15
> **Author Feedbacks to Reviewer kp2m (Part 1/2)**
>
> Thanks for your golden comments and constructive suggestions! We respond to your concerns as below:
>
> >**Q1: It is not intuitive to combine diffusion modeling with contrast-based learning.**
>
> Below we clarify the intuitions of **combining diffusion modeling and contrast-based learning for better representation learning ability**, where the claims are supported by empirical results.
>
> For protein representation learning, diffusion modeling and contrast-based methods show their strengths in different aspects: (1) **Diffusion modeling excels at modeling the dependencies within a single protein**, in which detailed atom- and residue-level interactions can be captured by protein structure diffusion, and residue type dependencies along protein sequences can be captured by protein sequence diffusion; (2) By comparison, **contrast-based methods are good at aligning representations from similar proteins**, where the representations of correlated proteins are constrained to be nearby in the latent space. By combining these orthogonal objectives of representation learning, **the learned model is expected to produce both (1) residue/atom interaction-aware representations and (2) structural similarity-aware representations.** Such merits cannot be attained by performing only diffusion modeling or contrast-based learning.
>
> **This claim is well supported by our ablation studies.** In Table 1 and 2, by comparing the performance of SiamDiff with its ablation variants without either diffusion modeling (the first line of last block) or contrast-based learning (the last line of last block), we can observe clear performance gains of SiamDiff which combines the two learning mechanisms.
>
> >**Q2: I was wondering is there some theoretical analysis between diffusion modeling and the introduced random noise?**
>
> We would like to clarify that **the introduced random noise is only used to acquire a correlated view of the original protein for contrast-based learning, instead of triggering the diffusion process**.  We defend the theoretical soundness of such a structure perturbation scheme as below:
>
> 1. **The random structure perturbation is valid under the mutual information (MI) maximization framework.** Our method is on the research line of MI maximization based representation learning, whose core idea is to construct correlated paired data and maximize the MI between the representations of each correlated pair. In our work, we deem a native protein and its counterpart with random structure perturbation as a correlated pair. We constrain the perturbation to be with small scale, and thus **the perturbed counterpart lies in the same local minima of the potential energy surface [a] as the native protein**, which forms valid correlated pairs for MI maximization according to a previous study of 3D molecular representation learning [b].
>
> 2. **The structure perturbation is not a part of the diffusion process.** We deem the original and the perturbed proteins as two different initial states, and two multimodal diffusion trajectories are independently constructed from these two initial states. Therefore, the structure perturbation has no impact on the diffusion process.
>
> To summarize, **the proposed structure perturbation is independent from the diffusion process, and thus its theoretical soundness under MI maximization is enough for the soundness of the whole learning framework.**
>
> &emsp;
>
> [a] Exploring potential energy surfaces for chemical reactions: an overview of some practical methods. Schlegel, H. Bernhard, Journal of computational chemistry, 2003.
>
> [b] Molecular geometry pretraining with se(3)-invariant denoising distance matching. Liu et al., arXiv:2206.13602 (2022).

---

> > ### Author Response · Authors · 2022-11-15
> > **Author Feedbacks to Reviewer kp2m (Part 2/2)**
> >
> > >**Q3: I do not quite agree with the claim that the added random noise could reflect the conformational fluctuation.**
> >
> > Appreciate for this insightful thought! We admit that the structure perturbation scheme used in our current method is not optimal for reflecting diverse structural change patterns of proteins. However, **this scheme well fits the mutual information (MI) maximization framework used in our method**. The employed structure perturbation is based on Gaussian noises with small scales, which, at most of the time, **makes the perturbed protein lie in the same local minima of the potential energy surface [a] as the original protein**. In this way, the original and the perturbed proteins can form valid correlated pairs for MI maximization, as illustrated in a previous study of 3D molecular representation learning [b].
> >
> > As you suggested, the patterns of protein conformational change can always be more complex than our used one. For example, the change of torsion angles on the protein backbone or side chains can always lead to drastic conformational change, which is not covered by our structure perturbation scheme. **The study of how to enhance protein structure representation learning with diverse types of protein conformational changes definitely needs more exploration and deserves an individual research project.** We leave this important study as one of our major future works.
> >
> > &emsp;
> >
> > [a] Exploring potential energy surfaces for chemical reactions: an overview of some practical methods. Schlegel, H. Bernhard, Journal of computational chemistry, 2003.
> >
> > [b] Molecular geometry pretraining with se(3)-invariant denoising distance matching. Liu et al., arXiv:2206.13602 (2022).

---

### Official Review · Reviewer_yEUV · 2022-10-29

**Confidence:** 2
**Correctness:** 4
**Technical Novelty And Significance:** 3
**Empirical Novelty And Significance:** 3
**Recommendation:** 5

**Clarity, Quality, Novelty And Reproducibility:**

- The method is novel and the methods derived in the paper are interesting.
- The writing is dense and assumes strong domain knowledge in proteins and is not really accessible to the general ML audience. Multiple concepts are not defined and this reviewer would find it hard to reproduce the work. The theoretical proofs are complete and fully reproducible to the point that results from cited papers are re-proved for convenience.

**Strength And Weaknesses:**

### Strengths
- The paper comes up with a lower bound for the mutual information between two diffusion sequences and uses it to lean an encoder for protein structure. This is an interesting case of theory guiding practice and makes the work interesting.
- The ablation study shows that all the components of the procedure contribute to improving the model.

### Weaknesses
- What are correlated views? These show up in the abstract and introduction without a simplified explanation and throw off a reader who is (even very slightly) outside the field.
- Table 1 and 2 are not color blind friendly, the community seems to use bold and underline to denote first and second, One can also use markers like $\dagger$ and $\circ$. Standard deviations are missing. Provide context for $F_{\text{max}}$, and $\rho$ in the caption for ease of skimming. In fact I was unable to find a definition of $F_{\text{max}}$.
- The abstract talks about comparing with motif based methods, this comparison doesn't show up in the experiments.
- What do you mean by `We do not include the results of GVP on EC and GearNet-Edge on RES tasks due to their poor performance.`?
- [Personal nitpick] Page 2 first sentence - shortening Tables to Tabs is non-standard.
- The writing was pretty dense and I required multiple reads to understand the text. However the figures helped and the fact that this is not my primary field of research is not the authors' fault.


**Summary Of The Paper:**

The authors consider the task of unsupervised learning of protein structures. They do so by maximizing the mutual information between two diffusion trajectories that start with a protein and a perturbed copy of the protein. The authors prove a lower bound of the mutual information which resembles an ELBO and specifically depends on the likelihood of one trajectory given the other.
As such one trajectory is used to compute the noises in the other trajectory.
By doing so, the procedure encodes both the structure and sequences jointly.

**Summary Of The Review:**

- I am giving this paper a weak reject due to the minor concerns noted in the above two sections, specifically about reporting on the experimental results and making the writing more accessible to the general audience at ICLR. I am willing to upgrade my rating based on my interaction with the authors during the discussion phase.

---

> ### Author Response · Authors · 2022-11-15
> **Author Feedbacks to Reviewer yEUV**
>
> Thanks for your insightful comments and valuable suggestions! We respond to your concerns as below:
>
> >**Q1: The introduction of “correlated views” lacks a simplified explanation.**
>
> In our paper, we introduce **correlated views** under the mutual information maximization framework, and they denote **different descriptions of the same protein** under such a context. In the revised version, we have clarified this point in the Abstract and in the second paragraph of Introduction.
>
> >**Q2: Table 1 and 2 are not color blind friendly.**
>
> Thanks for reminding us of this important point! In the Table 1 and 2 of the revision, we have removed the color-based annotations and used *, $^{\dagger}$ and $^{\ddagger}$ to denote the first, second and third best performance.
>
> >**Q3: Standard deviations are missed.**
>
> Thanks for reminding us! We admit that standard deviation is important. However, it is infeasible for us to finish these numerous experiments during rebuttal ($n \times 12 \times 6$ experiments in total, where $n$ is the number of repeated experiments). We’ll continue to work on this and report the numbers in the final version.
>
> >**Q4: The definitions of $F_{max}$, global $\rho$ and mean $\rho$ metrics should be provided for the ease of skimming.**
>
> This suggestion is great! In the revised version, we have supplemented the detailed definitions of $F_{max}$, global $\rho$ and mean $\rho$ to the Appendix D.
>
> >**Q5: The abstract talks about comparing with motif based methods, while this comparison doesn't show up in the experiments.**
>
> Previous contrastive-based protein structure pre-training methods seek to capture the co-occurrence of structure motifs in the same protein structure. These methods adopt random cropping functions to approximately sample structure motifs for contrastive learning, while **a pre-training method based on exact (not approximated) structure motifs have not been studied yet**. In our experiments, we have compared with such an approximated motif-based method, i.e., Multiview Contrast [a], and our proposed SiamDiff outperforms this baseline on both residue-level and atom-level benchmark tasks.
>
> [a] Protein representation learning by geometric structure pretraining. Zhang et al., arXiv:2203.06125 (2022).
>
> >**Q6: What do you mean by “We do not include the results of GVP on EC and GearNet-Edge on RES tasks due to their poor performance”?**
>
> Thanks for the question. GVP can only achieve 0.489 for F_max on EC as reported in [a], which is much lower than the performance of GearNet-Edge. In our preliminary experiments, GearNet-Edge can only achieve 0.290 for accuracy on RES. This is because GearNet-Edge is designed for modeling whole protein structures rather than small local structures around a residue in RES. Hence, we do not include the pre-training and fine-tuning results for GVP on EC and GearNet-Edge on RES.
>
> [a] Protein representation learning by geometric structure pretraining. Zhang et al., arXiv:2203.06125 (2022).
>
> >**Q7: Page 2 first sentence - shortening Tables to Tabs is non-standard.**
>
> Thanks for pointing this out! In the revision, we have substituted “Tab” with “Table” throughout the whole paper.
>
> >**Q8: The writing is pretty dense, which makes the paper not really accessible to the general ML audience.**
>
> Thanks for the valuable feedback! To better motivate the audience from the general community and make the main idea of our method more accessible, in the revised version, **we add two paragraphs at the beginning of Section 4.1 to better clarify (1) two main limitations of existing protein structure pre-training methods based on mutual information maximization, and (2) how the proposed multimodal diffusion trajectory can help to tackle these limitations.** We hope these contents can make our starting point and main idea more accessible. Please check them out!

---

### Author Response · Authors · 2022-11-15
**Summary of Response**

We would like to thank all reviewers for insightful comments and constructive suggestions!

We are working on the variance control for experimental comparisons - we will revise our paper to include standard deviations and significance tests as soon as possible. We have posted the responses to your questions and revised the paper for better readability, where **the revisions are marked in RED**. Here is a brief summary of important points:

1. **Readability of the paper (Reviewer yEUV, jQVb):** In the Abstract and Introduction of the revised paper, we have defined “correlated views” in a more comprehensible way. At the beginning of Section 4.1 of the revision, we have added two paragraphs to more clearly state the motivations of our method (i.e., two modeling limitations of previous methods and how our method can address the limitations).

2. **Technical soundness (Reviewer kp2m):** In the response, we analyze the complementarity of mutual information (MI) maximization and diffusion modeling on protein representation learning, which supports their combination for more effective representations. Also, we illustrate that the proposed structure perturbation scheme is simple yet well-suited to MI maximization and technically sound.

3. **Novelty concern (Reviewer UBXw):** In the response, we point out two main novel features of our method: (1) incorporating the idea of diffusion modeling into the MI maximization framework, which well captures the temporal information along protein formation; (2) combining continuous and discrete diffusion into a unified framework, which well models both protein structures and sequences.

---

### Decision · Program_Chairs · 2023-01-20

**Decision:**

Reject

**Justification For Why Not Higher Score:**

* Several reviewers pointed out that the method could be more intuitive. As diffusion models gradually apply noise to the input, introducing another level of random noise for contrast learning is not intuitive. The marginal performance increase also suggests conflict between the diffusion model and the contrast learning.
* There was a consensus among the reviewers that the paper needs more clarity. The notation feels overloaded, and it is hard to follow. In addition, multiple concepts are not defined.
* The code is currently not available. Several reviewers raised concerns that it could be hard to reproduce the work which was not addressed.
* Reviewers questioned the technical soundness of core method components. In particular, there is no clear linkage between the diffusion model and synthetic mutual information. Moreover, the added noise does not reflect changes in protein conformations.

**Justification For Why Not Lower Score:**

N/A

**Metareview: Summary, Strengths And Weaknesses:**

This paper considers the task of unsupervised learning of protein structures. This is achieved by applying a diffusion model to learn the protein structure from the protein sequence. The approach maximizes the mutual information between two diffusion trajectories that start with a protein and a perturbed copy of the protein and use one trajectory to compute the noise in the other trajectory. The paper provides a lower bound of the mutual information, which resembles an ELBO and depends explicitly on the likelihood of one trajectory given the other.

**Strengths:**
* Reviewers acknowledged some novel ideas relevant to protein representation learning. For example, the new method tried to achieve better protein residual presentation by introducing contrast learning and maximizing mutual information. However, the method showed only a marginal performance increase compared with existing methods.
*  The evaluation of the learned representations followed well-established benchmarks. Although the performance is not the highest for every task, some value of these representations is evident from the empirical results.


**Weaknesses:**
* Several reviewers pointed out that the method could be more intuitive. As diffusion models gradually apply noise to the input, introducing another level of random noise for contrast learning is not intuitive. The marginal performance increase also suggests conflict between the diffusion model and the contrast learning.
* There was a consensus among the reviewers that the paper needs more clarity. The notation feels overloaded, and it is hard to follow. In addition, multiple concepts are not defined.
* The code is currently not available. Several reviewers raised concerns that it could be hard to reproduce the work which was not addressed.
* Reviewers questioned the technical soundness of core method components. In particular, there is no clear linkage between the diffusion model and synthetic mutual information. Moreover, the added noise does not reflect changes in protein conformations.